# UltraCUA: Scaling Computer Use Agent through GUI and Programmatic Control

## Abstract

Multi-modal agents for computer use rely exclusively on primitive actions (click, type, scroll) that require accurate visual grounding and lengthy execution chains, leading to cascading failures and performance bottlenecks. While other agents leverage rich programmatic interfaces (APIs, MCP servers, tools), computer-use agents (CUAs) remain isolated from these capabilities. We present UltraCUA, a foundation model that bridges this gap through hybrid control—seamlessly integrating GUI primitives with high-level programmatic tool calls. To achieve this, our approach comprises four key components: (1) an automated pipeline that scales programmatic tools from software documentation, open-source repositories, and code generation; (2) a synthetic data engine producing 17,000+ verifiable tasks spanning real-world computer-use scenarios; (3) a multi-agent system generating high-quality hybrid control trajectories with both low-level GUI actions and high-level programmatic tool calls; and (4) a two-stage training pipeline combining supervised fine-tuning with online reinforcement learning, enabling strategic alternation between low-level and high-level actions. Experiments with our 7B and 32B models demonstrate substantial improvements over state-of-the-art agents. On OSWorld, UltraCUA models achieve an average 27% relative improvement over base models, while being 11% faster in terms of steps. Out-of-domain evaluation on WindowsAgentArena shows our model reaches 21.7% success rate, outperforming baselines trained on Windows data. The hybrid control mechanism proves critical, reducing error propagation while maintaining execution efficiency. This work establishes a scalable paradigm that bridges primitive GUI interactions and programmatic intelligence for stronger and unified computer use.

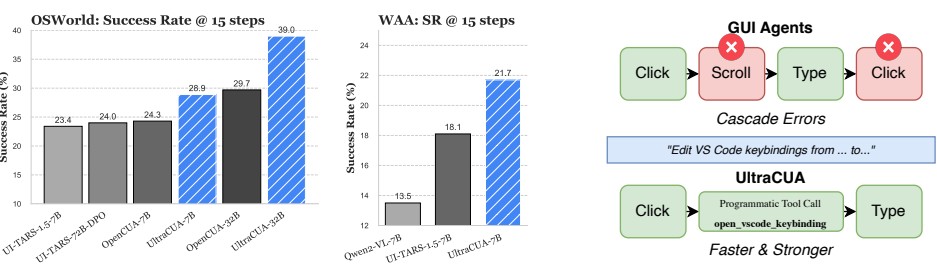

| (a) OSWorld | (b) WindowsAgentArena | (c) GUI Agents v.s. UltraCUA |

Figure 1: (a)(b): UltraCUA's performances; (c) Comparison between GUI Agents and UltraCUA.

## 1 Introduction

Computer-use automation has emerged as a critical capability for enabling autonomous agents to interact with the vast ecosystem of desktop and web applications that humans use daily (Hong et al., 2023; Shaw et al., 2024; Zhang et al., 2023). However, current computer-use agents (CUAs) face a fundamental limitation: they operate exclusively through primitive actions such as clicking, typing, and scrolling (Rawles et al., 2024; Koh et al., 2024). This constraint creates a significant performance gap compared to agents that leverage rich programmatic interfaces—APIs, MCP servers, and tools—to accomplish complex tasks efficiently (Qin et al., 2023b; Schick et al., 2023b).

The reliance on primitive actions introduces critical challenges. First, lengthy execution chains accumulate errors that cascade into failures—a single misplaced click can derail an entire task (Zheng

et al., 2024; Yan et al., 2023). Second, operations that could be accomplished with a single programmatic call require dozens of GUI actions, creating performance bottlenecks. For example, extracting data from multiple spreadsheets requires a traditional CUA to navigate menus, select cells individually, copy values, switch applications, and paste content—each action a potential failure point. In contrast, an agent with spreadsheet APIs could accomplish this reliably with far fewer operations. This efficiency gap is stark: while other agents leveraging programmatic interfaces exceed 80% success on benchmarks like GAIA (Mialon et al., 2024; Zhang et al., 2025), GUI-only computer-use agents remain fundamentally limited, motivating our unified approach that combines GUI generality with programmatic efficiency.

In this paper, we bridge this capability gap through **hybrid control**, seamlessly integrating GUI primitives with high-level programmatic tool calls. Rather than treating these as mutually exclusive options, our approach enables a strategic combination of both modes. Agents learn to leverage programmatic tool calls when they provide clear efficiency gains, while retaining GUI interactions for universal coverage and fine-grained control. To summarize, our technical contributions include:

- **An automated pipeline for collecting programmatic tools** that scales beyond manually curated sets (Qin et al., 2023a; Tang et al., 2023). Our system extracts tools from software documentation, integrates open-sourced implementations, and employs coding agents to generate new tools on demand. This scalable pipeline produces hundreds of tools across diverse environments, from OSWorld's Ubuntu applications to WindowsAgentArena's Windows ecosystem.

- **A dual-pipeline synthetic data engine** for verifiable computer-use task generation. Large-scale task synthesis for CUA training is challenging due to the complexity of verifying task completion in dynamic environments. To address this, we develop two complementary pipelines producing 17,000+ verified tasks. The first pipeline employs an instruction-first strategy where agents explore computer environments and propose tasks based on observed states, with trajectories verified by evaluator agents. The second pipeline uses an evaluator-first strategy, collecting atomic verification functions (e.g., checking Chrome URLs, verifying file paths, validating image attributes) from environments, then reprogramming (e.g., modifying parameters) and composing (e.g., combining multiple checks) them to create complex evaluation criteria. LLMs generate tasks satisfying these pre-defined evaluators, ensuring reliable trajectory assessment for training.

- **A large-scale hybrid control trajectory collection**. Existing computer-use datasets contain only pure GUI action sequences, lacking demonstrations of programmatic tool integration. We collect 20,000+ successful trajectories by combining a powerful planner model (OpenAI o3) with a state-of-the-art grounding model (GTA1-7B Yang et al. (2025))—a simple yet effective agentic framework. The planner selects between programmatic tool calls and low-level GUI actions based on task context, while the grounder ensures accurate GUI execution. This dataset enables training models to seamlessly alternate between action modes for optimal task completion.

- **A foundation agent model with hybrid control** trained using the programmatic tools, synthetic tasks, and rollout trajectories described above. We train models at two scales (7B and 32B) through supervised fine-tuning on the high-quality trajectories from our collection, followed by online reinforcement learning on our verifiable synthetic tasks. This two-stage approach produces agents that effectively select between GUI primitives and programmatic tool calls based on task context.

Experiments demonstrate substantial improvements over state-of-the-art CUAs. On OSWorld (Xie et al., 2024), our models achieve an average 27% relative improvement over their base models across both scales. Notably, out-of-domain evaluation on WindowsAgentArena Bonatti et al. (2024)—without any Windows-specific training—shows our 7B model reaches 21.7% success rate, outperforming baselines trained on Windows data. These results validate that hybrid control provides consistent benefits across model scales and platforms. Our code, models, and datasets will be released to facilitate future research.

## 2 METHODOLOGY

Our methodology comprises three key components for developing a foundation CUA model with hybrid control. First, we build a comprehensive collection of programmatic tools through an automated extraction pipeline. Second, we design a dual-pipeline synthetic data engine that generates verifiable tasks for complex real-world computer use. Finally, we train our model via supervised fine-tuning on collected trajectories followed by online reinforcement learning on synthetic tasks.

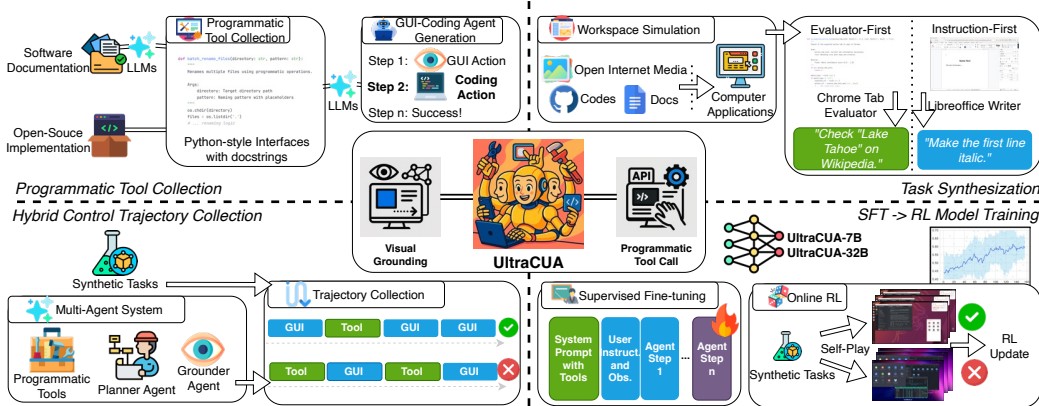

Figure 2: An overview of UltraCUA's design.

## 2.1 AUTOMATED TOOL COLLECTION FOR HYBRID CONTROL

The foundation of our approach is **hybrid control**—seamlessly integrating primitive GUI actions with high-level programmatic tools. We define a "tool" as a high-level interface encapsulating sequences of computer-use actions, typically implemented as Python functions, keyboard shortcuts, or combinations of primitive actions (e.g., type, key combinations)—but excluding actions requiring visual grounding like clicks. Each tool is exposed to the model through a Python function signature with descriptive docstrings specifying parameters and functionality.

While GUI-only agents suffer from cascading failures in lengthy action sequences, programmatic interfaces alone cannot handle all computer interactions. Our hybrid approach enables agents to leverage programmatic tools for efficiency when available, while retaining GUI actions to ensure generalization. To build a hybrid action space where tools cover diverse applications and usage scenarios, we developed an automated pipeline collecting hundreds of tools from the following three complementary sources. Tool details are also present in Appendix A.3.

**Extraction from Software Documentation.** Application documentation contains expert knowledge—particularly keyboard shortcuts—that bypass tedious GUI sequences. For example, changing VS Code's color theme requires navigating File → Preferences → Color Theme with GUI actions. Our pipeline extracts the shortcut (Ctrl+K, Ctrl+T) from documentation and converts it into a programmatic tool: vscode.set_theme(). This transforms fragile multi-step sequences into single, reliable operations.

**Integration of Open-Source Implementations.** We incorporate existing programmatic tools from open-sourced frameworks, particularly leveraging implementations from AgentS2 (Agashe et al., 2025) and AgentStore (Jia et al., 2024). These tools transform complex GUI sequences into efficient programmatic calls. For example, this AgentS2 tool for spreadsheet manipulation replaces dozens of manual clicks with a single function:

```python
def set_cell_values(self, cell_values: dict, app_name: str, sheet_name:
    str):
    """Set multiple cell values in a spreadsheet.
    Args: cell_values: {"A2": "hello", "B3": 123.45}"""
    return SET_CELL_VALUES_CMD.format(
        cell_values=cell_values, app_name=app_name, sheet_name=sheet_name
    )
```

**Automatic Scaling with Coding Agents.** Inspired by CoACT-1 (Song et al., 2025), we adopt the multi-agent paradigm where an orchestrator dynamically delegates subtasks to either a GUI operator or a coding agent that executes Python/Bash scripts. This allows bypassing inefficient GUI sequences through direct programmatic execution. We extend this by mining the coding agent's trajectories for reusable tools: when the coding agent solves subtasks programmatically, we employ an automatic LLM workflow to extract and refine these solutions into parameterized functions, with reflection steps and automated unit testing to ensure correctness. For example, from a trajectory where the coding agent modifies VS Code settings via script, we extract:

```
def add_vs_code_keybinding(key: str, command: str, when: str = ""):
    """Create or update a VS Code keybinding.
    Args: key: "ctrl+j", command: "workbench.action.
    focusActiveEditorGroup"
    Returns: {"path": "...", "action": "added", "backup": "..."}"""
```

## 2.2 SYNTHETIC DATA ENGINE FOR HYBRID CONTROL TASKS

Large-scale synthetic training tasks for CUAs remains scarce, while existing resources are primarily test sets or complete trajectories with limited reproducibility. To address this, we developed a **dual-pipeline synthetic data engine** producing 17,000+ verifiable tasks for real-world computer-use. Our engine operates through two complementary strategies: **evaluator-first** generation ensuring verifiability and **instruction-first** generation creating contextually relevant tasks with diversity.

### 2.2.1 EVALUATOR-FIRST GENERATION

This approach begins by collecting state-checking evaluators from computer environments—scripts that verify specific system states (e.g., file existence, application settings, UI elements). We use the atomic evaluator functions in OSWorld (Xie et al., 2024) to reprogram these evaluators by modifying parameters and compose multiple evaluators to create complex verification conditions. For example, combining a file-checker with a URL-checker validator creates a task requiring both file manipulation and browsing interaction.

Given these evaluator configurations, we prompt LLMs to generate corresponding tasks that would satisfy the verification conditions. For instance, the file-URL checker combination might generate tasks like "Navigate to the Python documentation page and download the PDF tutorial to your Documents folder," which requires both web browsing to reach the correct URL and file system operations to verify the download. This ensures every generated task has a programmatic way to verify completion, critical for providing clear reward signals during RL training. This approach produces 4,000+ high-quality tasks with guaranteed verifiability.

### 2.2.2 INSTRUCTION-FIRST GENERATION

Following Anonymous (2025), this approach generates tasks based on observed system states. Agents explore computer environments through exploratory walks, reaching diverse UI states. At each state, we analyze the current interface and generate contextually appropriate tasks (e.g., "create a new spreadsheet" when in a file manager). Task completion is verified by an evaluator agent rather than predefined scripts, allowing flexibility in execution paths. This approach generates 12,000+ tasks that naturally arise from real usage patterns, complementing the systematic coverage of evaluator-first generation.

### 2.2.3 WORKSPACE SIMULATION

A realistic workspace is crucial for generating meaningful tasks. When synthetic tasks require interaction with specific content, our pipeline triggers a content preparation workflow tailored to task requirements. For example, for code-related tasks, we fetch files from popular GitHub repositories—extracting Python scripts from Hugging Face repos or configuration files from trending projects. For image tasks, we retrieve open-source images from Wikipedia Commons matching relevant categories. For document editing, we generate synthetic documents via LLMs with task-appropriate content. This targeted approach ensures realistic task contexts: image editing tasks receive actual photos, code refactoring tasks get real implementations, and document tasks operate on properly formatted files. By matching content types to task requirements, we create scenarios that accurately reflect real-world computer use.

### 2.2.4 COMPLEMENTARY DESIGN RATIONALE

In general, the two approaches serve distinct purposes. Evaluator-first generation produces complex, verifiable tasks ideal for RL training—code-based evaluators provide precise rewards without expensive trajectory verification. However, these tasks tend to be challenging due to evaluators' design and multi-evaluator compositions. Instruction-first generation offers greater diversity through environment exploration, covering more real-world scenarios with naturally easier tasks. This comple-

mentary design ensures both reliable RL signals and broad task coverage. We further have detailed data statistics in Table 6.

### 2.3 Training a Foundation Agent for Hybrid Control

We train our foundation model using a two-stage approach: supervised fine-tuning on high-quality trajectory demonstrations followed by online reinforcement learning. This curriculum first establishes competency in hybrid control, then optimizes action selection between GUI primitives and programmatic interfaces through self-play.

#### 2.3.1 Multi-Agent Rollout for Trajectory Generation

To generate high-quality training data, we deploy a multi-agent system comprising a Planner agent and a specialized Grounder agent. We use OpenAI o3 as the Planner, which operates in a ReAct framework (Yao et al., 2022) with Agent-S2-style prompting (Agashe et al., 2025) to enhance reasoning capabilities. The Planner strategically chooses between programmatic calls and GUI actions based on task context and available tools. When GUI interaction is needed, we employ GTA1-7B (Yang et al., 2025) as the Grounding agent for precise visual localization, ensuring accurate element targeting in complex interfaces. For each synthetic task, we expose relevant programmatic tools to the Planner and perform 8 rollouts to capture diverse solution strategies. This process generates 26.8K successful trajectories demonstrating effective hybrid control strategies across our synthetic tasks.

#### 2.3.2 Working Memory Mechanism

Complex hybrid execution paths risk losing context as agents alternate between programmatic tools and GUI actions. We address this through an integrated working memory system using `<memory></memory>` tags, inspired by Bonatti et al. (2024). The agent autonomously maintains this memory—recording completed steps, extracted values, and intermediate results—ensuring coherent execution without external storage. The common memory content includes: (1) task objectives and constraints, (2) progress tracking across completed actions, and (3) information that must persist across steps (e.g., file paths, UI element states, intermediate values). For example, during a bookmark management task, the agent maintains structured state information as shown. This mechanism proves crucial for multi-step tasks requiring information persistence across action modality switches.

```
<memory>
Task: Create folder 'Favorites' on
    bookmarks bar.
Progress: Chrome open, bookmarks bar
    visible.
Next: Access bookmark manager via
    Ctrl+Shift+O.
</memory>
```

#### 2.3.3 Stage 1: Supervised Fine-Tuning

We fine-tune multiple base models, including UI-TARS-1.5 (7B) (Qin et al., 2025) and OpenCUA (32B) (Wang et al., 2025b) on the 26.8K successful trajectories from the rollout system. To ensure balanced training across all trajectory steps, we create individual samples from each turn: for the $i$-th turn, we include messages up to that point but apply loss only to the $i$-th assistant response. This prevents overfitting to early trajectory steps while ensuring each action decision receives equal training weight, teaching the model proper hybrid control at every step of task execution.

#### 2.3.4 Stage 2: Online Reinforcement Learning

While SFT provides behavioral foundations, mastering strategic action selection requires learning from exploration. The hybrid action space creates numerous solution paths for each task—some efficient, others suboptimal. Through online RL, agents can discover these optimal strategies via self-play.

We begin by filtering our evaluator-first tasks (4,000+) through 8 rollouts per task with the SFT model, identifying 1,000 tasks where the model succeeds at least once. We define task difficulty as the average success rate across these 8 rollouts. During training, we randomly sample tasks

with difficulty scores in [0.4, 0.8]—avoiding tasks that are too easy or too challenging to maximize learning efficiency within the model's zone of proximal development.

For policy optimization, we employ a variant of GRPO (Shao et al., 2024) inspired by DAPO (Rosset et al., 2024), with key modifications for our hybrid control setting. We remove KL regularization and implement a clip-higher strategy to encourage exploration of diverse action sequences.

To prevent regression toward GUI-only solutions, we design a reward function that incentivizes tool usage. The total reward for a trajectory $\tau$ is:

$$R(\tau) = R_{\text{env}}(\tau) + R_{\text{tool}}(\tau) \tag{1}$$

where $R_{\text{env}}(\tau) \in \{-1, 1\}$ is the sparse environment reward (1 for task success, -1 for failure), and the tool-use reward is defined as:

$$R_{\text{tool}}(\tau) = \begin{cases} 0.3 & \text{if } R_{\text{env}}(\tau) = 1 \text{ and } \tau \text{ contains tool calls} \\ 0 & \text{otherwise} \end{cases} \tag{2}$$

This reward structure teaches the agent not just to succeed, but to succeed efficiently through strategic hybrid control. Notably, we exclude format rewards despite their common use in RL with LLMs. We found in empirical analysis that models struggle with complex tool syntax early in training, causing format penalties to dominate the learning signal and discourage outcome-based learning. By focusing solely on outcome and tool-use rewards, we enable the model to gradually master tool syntax through successful examples rather than punishment, leading to more robust learning. We propagate rewards to each action step and normalize by trajectory length for stable optimization.

## 3 EXPERIMENTS

### 3.1 EXPERIMENTAL SETUP

#### 3.1.1 BENCHMARKS

We use **OSWorld-Verified** (Xie et al., 2024) as our primary benchmark. It is a realistic benchmark featuring a Ubuntu Desktop environment accessible through screen observations, comprising 369 tasks. OSWorld contains diverse tasks spanning common office suites, IDEs, and web browsers, designed to rigorously test an agent's long-horizon planning and visual grounding abilities. Each task is self-contained with a deterministic starting state, a natural language goal, and an automated rule-based evaluator, ensuring reproducible and reliable assessment. To evaluate cross-platform generalization, we also test on **WindowsAgentArena** (Bonatti et al., 2024), which contains 154 real-world tasks in Windows 11 environments. This provides an out-of-domain evaluation since our models are primarily trained on Ubuntu-based tasks, testing the transferability of learned hybrid control strategies across operating systems.

#### 3.1.2 BASELINES

To demonstrate the effectiveness of our approach, we compare our final model against several strong baselines that isolate different components of the agent's capabilities.

- **General Models:** powerful, pre-trained vision-language models that are not specifically fine-tuned for GUI automation. We include leading models like Claude (Anthropic, 2025) and o3 (OpenAI, 2025) to establish a baseline for generalist, out-of-the-box performance.

- **Multi-Agent Frameworks:** systems that orchestrate multiple components to solve computer-use tasks. These frameworks typically employ a planner-grounder architecture and may be enhanced with additional capabilities such as memory, experience replay, or the integration of a coding agent. Prominent examples include Agent-S2 (Agashe et al., 2025) and Jedi-7B (Xie et al., 2025).

- **Specialized Agentic Models:** models that have been specifically fine-tuned or purpose-built for computer control and GUI-centric scenarios. This includes models like OpenAI CUA OpenAI (2025) UI-TARS (Qin et al., 2025) and OpenCUA (Wang et al., 2025b), which are trained on large datasets of computer interaction trajectories to specialize their abilities for this domain.

Table 1: Comparison of the state-of-the-art methods on the OSWorld benchmark. We split the results by steps and show the approach type in the second column. We report the success rate (%) as the evaluation metric in the fourth column. † denotes our reproduced results, averaged across 4 independent runs. Same-colored rows share the same base model.

| Agent Method | Model Category | Open-Source | Success Rate (%) | |
| --- | --- | --- | --- | --- |
| | | | Max Steps: 15 | Max Steps: 50 |
| o3 (OpenAI, 2025) | General Model | ✗ | 9.1 | 17.2 |
| Claude 3.7 Sonnet (Anthropic, 2025) | General Model | ✗ | 27.1 | 35.8 |
| OpenAI CUA (OpenAI, 2025) | Agentic Model | ✗ | 26.0 | 31.3 |
| Jedi-7B w/ GPT-4o (Xie et al., 2025) | Multi-Agent Framework | ✓ | 26.8 | 27.0 |
| Agent S2 (Agashe et al., 2025) | Multi-Agent Framework | ✓ | 27.0 | 34.5 |
| Qwen2.5-VL-72B (Bai et al., 2025) | General Model | ✓ | 4.4 | – |
| UI-TARS-72B-DPO (Qin et al., 2025) | Agentic Model | ✓ | 24.0 | 25.8 |
| OpenCUA-7B (Wang et al., 2025b) | Agentic Model | ✓ | 24.3 | 28.2 |
| UI-TARS-1.5-7B (Qin et al., 2025) | Agentic Model | ✓ | 23.4† | 26.1† |
| OpenCUA-32B (Wang et al., 2025b) | Agentic Model | ✓ | 29.7 | 34.1 |
| **UltraCUA-7B** | Agentic Model | ✓ | **28.9†** | **30.2†** |
| **UltraCUA-32B** | Agentic Model | ✓ | **39.0†** | **41.5†** |

### 3.1.3 IMPLEMENTATION DETAILS

**Training Details.** Our models are fine-tuned for 3 epochs during the SFT stage with a learning rate of 2e-5. For SFT stage, we sample 66K steps from trajectories with evaluator-first and instruction-first synthetic data, each 33K. The subsequent online RL stage is trained for 150 steps with a learning rate of 1e-6. All experiments are conducted on NVIDIA H100 GPUs. During training, we control the number of programmatic tools to limit the context length at 32K.

**Evaluation Metrics.** We use the following metrics to measure effectiveness and efficiency: **1) Success Rate (SR):** Our primary metric. It is the percentage of tasks the agent successfully completes in a single attempt, as verified by the benchmark's automated evaluators. **2) Pass@4:** To account for the stochastic nature of LLM inference, we also report Pass@4. A task is marked as successful under this metric if the agent completes it correctly in at least one of four independent rollout attempts. **3) Trajectory Efficiency:** We measure the number of steps an agent takes to successfully complete a task. Each step is either a GUI action or a programmatic tool call. A lower step count indicates higher efficiency.

## 3.2 MAIN RESULTS

**OSWorld Evaluation.** Table 1 presents comprehensive results on the OSWorld benchmark across different step budgets. Our UltraCUA-7B achieves 28.9% success rate at 15 steps, surpassing all comparable 7B models including the strong UI-TARS-1.5-7B baseline (23.4%) with a 23.5% relative improvement. More remarkably, UltraCUA-32B reaches 39.0% success rate, outperforming even closed-source systems like Claude 3.7 Sonnet (27.1%) and OpenAI CUA (26.0%).

The results validate our hybrid control approach across model scales. While general-purpose models struggle without specialized training (e.g., Qwen2.5-VL-72B at 4.4% despite 72B parameters), our models achieve superior performance through strategic integration of programmatic tool calls. The consistent improvements from base models (UI-TARS-1.5-7B→UltraCUA-7B: +23.5%, OpenCUA-32B→UltraCUA-32B: +31.3%) demonstrate that hybrid control provides orthogonal benefits to agent capabilities.

**Cross-Platform Generalization.** To assess generalization beyond the training domain, we evaluate on WindowsAgentArena without any Windows-specific fine-tuning. Table 2 shows that UltraCUA-7B achieves 21.7% success rate, outperforming both Qwen2-VL-7B trained with OpenCUA's Windows data (13.5%) and UI-TARS-1.5-7B (18.1%). This 20% relative improvement over UI-

Table 2: Out-of-domain evaluation on WindowsAgentArena.

| Model | SR (%) |
| --- | --- |
| Qwen2-VL-7B (w/ OpenCUA Data) | 13.5 |
| UI-TARS-1.5-7B | 18.1 |
| **UltraCUA-7B** | **21.7** |

TARS demonstrates that hybrid control strategies learned on Ubuntu effectively transfer to Windows environments, validating the domain-agnostic nature of our approach.

### 3.3 ABLATION STUDIES

We conduct a series of ablation studies to dissect our framework and validate the contribution of its key components. These experiments isolate the impact of the hybrid action space, working memory, and reinforcement learning stage on agent performance.

#### 3.3.1 THE IMPACT OF HYBRID CONTROL

To validate the effectiveness of hybrid control, we examine its impact on both specialized agentic models and powerful multi-agent frameworks.

**Impact on Specialized Models.** We compare three configurations: (1) UI-TARS-1.5-7B (GUI-only baseline), (2) our model with tools disabled (UltraCUA-7B w/o Tools), and (3) our full model with hybrid control. Table 3 shows that hybrid control yields substantial improvements: success rate increases from 21.8% to 27.0% (+23.9% relative) while maintaining similar step counts. The addition of programmatic tools proves essential for effectiveness in complex automation tasks.

**Impact on Multi-Agent Frameworks.** To test whether hybrid control benefits extend to state-of-the-art systems, we evaluate our GTA1-7B + o3 rollout framework with and without programmatic tools. As shown in Table 3, hybrid control provides even larger gains in this setting: success rate improves from 44.0% to 48.2% (+9.5% relative) and average steps decrease by 14.9%. This demonstrates that hybrid control becomes increasingly valuable as the underlying system becomes more capable.

Table 3: Impact of hybrid control on different agent architectures. Hybrid control benefits both specialized models and multi-agent frameworks.

| Model Configuration | Success Rate (%) | Pass@4 | Avg. Steps |
|---|---|---|---|
| *Agentic Models (Max Steps: 15)* | | | |
| UI-TARS-1.5-7B (GUI-Only) | 23.4 | 33.3 | 9.31 |
| UltraCUA-7B-SFT w/o Tools (GUI-Only) | 25.1 | 34.3 | 9.24 |
| UltraCUA-7B-SFT (Hybrid Control) | **27.0** | **37.9** | **8.46** |
| *Commercial Models & Multi-Agent Framework (Max Steps: 50)* | | | |
| Claude-4-Sonnet | 43.9 | – | – |
| GTA1-7B + o3 w/o Tools | 44.0 | 60.5 | 15.53 |
| GTA1-7B + o3 (Hybrid Control) | **48.2** | **62.4** | **13.22** |

#### 3.3.2 THE IMPORTANCE OF REINFORCEMENT LEARNING

We evaluate the impact of online RL by comparing models before and after this training stage, for UltraCUA-7B. From Table 1 (RL results) and Table 3 (SFT results), we can see that online RL brings 7% overall improvement (27.0→28.9). Figure 3 reveals how RL transforms agent behavior in three key ways. First, outcome rewards increase steadily during RL (Fig. 3a), confirming performance gains. Interestingly, format rewards also improve substantially (Fig. 3b) despite not being explicitly optimized. This suggests agents learn proper tool syntax naturally through successful task completion. Most significantly, RL reshapes tool-use strategy (Fig. 3c). Tool-related failures drop 46% (122→66) while successes increase by 5%, indicating pre-RL models often make harmful tool calls. Correspondingly, overall tool usage decreases, showing agents learn to be selective rather than aggressive with tool deployment. These results demonstrate that while SFT teaches the mechanics of hybrid control, RL enables strategic decision-making about *when* to use each action type—a crucial distinction for effective automation.

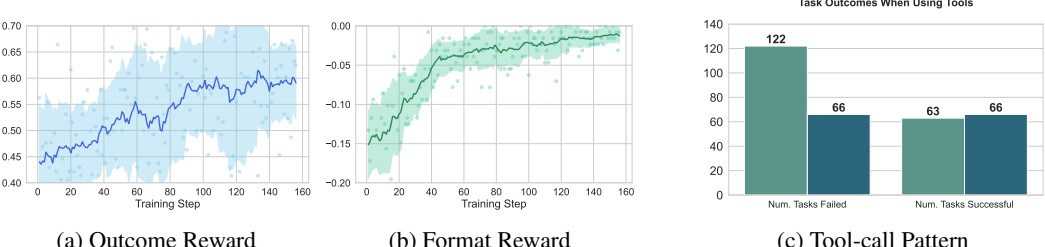

| (a) Outcome Reward | (b) Format Reward | (c) Tool-call Pattern |

Figure 3: Evolution of agent behavior during reinforcement learning.

#### 3.3.3 IMPACT OF WORKING MEMORY

We evaluate working memory by training models with and without `<memory></memory>` blocks in the SFT data, isolating the contribution of explicit state tracking. Table 4 shows consistent im-

provements from working memory: success rate increases from 25.4% to 27.0% (+6.3% relative) and average steps decrease slightly. While modest, these gains are meaningful for tasks requiring persistent state—file operations, form filling, and cross-application workflows. The efficiency improvement suggests memory helps agents avoid redundant actions like re-navigating to previously visited screens or re-extracting obtained information.

Table 4: Impact of working memory on model performance. Models are trained with identical data except for the presence of memory blocks.

| Model Configuration | Success Rate (%) | Pass@4 | Avg. Steps |
|---|---|---|---|
| UltraCUA-7B-SFT w/o Memory | 25.4 | 37.1 | 8.56 |
| UltraCUA-7B-SFT w/ Memory | **27.0** | **37.9** | **8.46** |
| *Relative Improvement* | **+6.3%** | **+2.1%** | **-1.2%** |

## 3.4 ANALYSIS

### 3.4.1 TOOL USAGE PATTERNS

To understand how our model leverages the hybrid action space, we analyze tool usage patterns across different application domains and task types.

**Tool Usage Scales with Model Capability.** Figure 4 reveals a clear correlation between model capability and tool usage sophistication. The multi-agent framework (GTA1-7B+o3) demonstrates extensive tool utilization with 60-80 calls and 8-10 unique tools per domain, while our single models show progressively conservative patterns—UltraCUA-32B uses tools moderately (20-40 calls) and UltraCUA-7B sparingly (0-20 calls). This pattern validates our hybrid control hypothesis: stronger models not only call tools more frequently but also leverage greater diver-

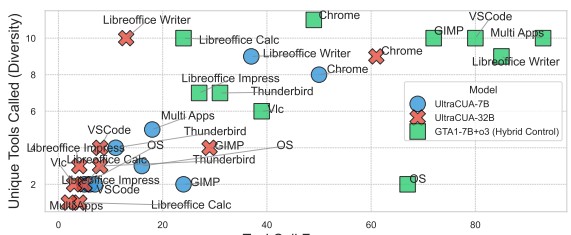

Figure 4: Tool-call patterns across domains and models. Stronger models exhibit higher frequency and diversity.

sity, suggesting they better recognize when programmatic interfaces provide efficiency gains. The trend holds across all domains from office suites to development environments, confirming that effective hybrid control emerges naturally with increased model capability.

**Out-of-Distribution Tool Generalization.** We evaluate the model's ability to utilize tools not seen during training by introducing new programmatic tools at inference time. These tools are unseen during training due to context length limit. Table 5 shows that models can adapt to unseen tools, achiev-

Table 5: OSWorld with OOD tools.

| Configuration | SR (%) | Avg. Steps |
|---|---|---|
| UltraCUA-7B-SFT | 27.0 | 8.46 |
| w/ OOD tools | **27.5** | 8.80 |

ing modest performance gains (+1.9% relative SR). However, the increased steps suggests adaptation challenges—models may explore unfamiliar tools before selecting appropriate ones. This zero-shot tool generalization capability also extends beyond single-platform scenarios: Table 2 demonstrates that our model achieves 21.7% success rate on Windows tasks despite training exclusively on Ubuntu, outperforming baselines by leveraging its learned hybrid control strategies across platforms and tool ecosystems.

## 4 CONCLUSION

We introduced UltraCUA, a foundation agent that bridges the critical gap between general-purpose GUI agents and specialized API-based agents. We achieve this through a novel hybrid action space that seamlessly integrates low-level GUI actions with high-level tool use. Our core contributions are a scalable pipeline for automated tool acquisition, a synthetic data engine for generating verifiable hybrid tasks, and a two-stage SFT+RL curriculum to teach strategic action selection. Our method achieves state-of-the-art performance on the OSWorld benchmark. Ablation studies confirm that the hybrid action space is the essential driver of this success, demonstrating a new and more effective paradigm for building robust and efficient agents for general computer control.

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

## A APPENDIX

### A.1 RELATED WORK

**Multi-Modal Agents for Computer Automation.** The ambition to create agents that can operate graphical user interfaces is long-standing, but has seen remarkable progress with the advent of Vision-Language Models (VLMs). Early approaches often relied on structured data like HTML or accessibility trees. More recent and generalizable agents operate directly from pixels and high-level instructions. In web automation, benchmarks like WebArena (Zhou et al., 2023) and Mind2Web (Deng et al., 2023) have driven the development of agents capable of complex online tasks. Similarly, in general computer control, works like CogAgent (Hong et al., 2023) and OSWorld (Xie et al., 2024) have demonstrated agents that can navigate desktop environments, and AppAgent (Zhang et al., 2023) has shown similar capabilities on mobile devices. Current approaches to GUI automation can be broadly categorized into two paradigms. Multi-Agent systems employ specialized models for different subtasks—for instance, GPT-4o+Aria-UI (Yang et al., 2024) and GTA-1 Yang et al. (2025) combine a planner model with a dedicated grounder model, leveraging the strengths of each component for strategic planning and precise visual grounding respectively. In contrast, Foundation Agent Models like UI-TARS (Qin et al., 2025), UI-TARS-2 (Wang et al., 2025a), and OpenCUA (Wang et al., 2025b) adopt an end-to-end approach, where a single unified model autonomously handles both planning and grounding tasks. While multi-agent systems benefit from modular design and specialized expertise, foundation models offer simpler deployment and potentially better coordination between planning and execution. A common thread among these powerful agents is their reliance on a primitive action space consisting of clicks, types, and scrolls. While this provides generality, it also leads to the brittleness and long-horizon planning challenges that our work directly addresses. Our contribution is the introduction of a hybrid action space that retains this generality while adding the efficiency and robustness of high-level tools.

**Tool and API Augmentation for LLMs.** Parallel to the development of GUI agents, another line of research has focused on augmenting Large Language Models (LLMs) with the ability to use external tools and APIs. The seminal work of ToolFormer (Schick et al., 2023a) showed that models could learn to call APIs to access information they lack. This paradigm was rapidly scaled up by frameworks like ToolLLM (Qin et al., 2023b) and the Gorilla benchmark (Patil et al., 2023), which enabled models to select from thousands of real-world APIs. Furthermore, the concept of "tool-making" (Cai et al., 2023) has explored agents that can write their own tools when needed, a capability we incorporate into our tool acquisition pipeline. Recent advances have introduced reinforcement learning to tool-use training. ReTool (Feng et al., 2025) and ToolRL (Qian et al., 2025) pioneered the use of online RL for training end-to-end tool-use agents, demonstrating that reward signals alone can guide models to learn effective tool selection and usage strategies. These methods move beyond supervised learning on static datasets, allowing agents to discover optimal tool-use patterns through interaction and feedback. This RL-based paradigm aligns closely with our approach, where we employ online reinforcement learning to train agents that can strategically alternate between primitive GUI actions and high-level tool calls. While these tool-augmented agents are highly effective for structured, programmatic tasks, they typically operate in a non-visual, text-based environment and lack the ability to interact with the vast number of applications that do not expose an API. Our work bridges this gap, bringing the power of a rich tool ecosystem to the visually-grounded domain of GUI agents.

Table 6: Comparison of our two synthetic data generation strategies.

| Synthesization Strategy | Task Count | Rollout SR (%) | Avg. Difficulty | Avg. Steps | Total Samples | Total Trajectories |
|---|---|---|---|---|---|---|
| Evaluator-First | 4K | 29 | Medium-Hard | 6.8 | 33K | 4.8K |
| Instruction-First | 13K | 45 | Easy-Medium | 6.5 | 149K | 22K |

### A.2 THE USE OF LARGE LANGUAGE MODELS

We used large language models (LLMs) to assist with specific aspects of paper preparation. Specifically, LLMs were employed for: (1) language polishing and grammar checking to improve clarity and readability, (2) formatting suggestions to ensure compliance with conference style guidelines, and (3) recommendations for data visualization approaches to better present experimental results. All research ideas, experimental design, implementation, and core scientific contributions were de-

veloped by the authors without LLM assistance. The LLMs served purely as writing and presentation aids.

### A.3 DETAILS FOR PROGRAMMATIC TOOLS

Table 7 summarizes the programmatic tools available across 10 different application domains on OSWorld. The collection comprises 881 tools in total, with individual domains offering between 4 (System) and 135 (VS Code) tools. These tools provide fine-grained control over desktop applications, enabling agents to perform tasks ranging from basic navigation (e.g., `jump_to_next_tab`) to complex application-specific operations (e.g., `batch_spreadsheet_numeric_formatter`). The comprehensive tool coverage ensures that agents can effectively automate diverse desktop workflows across different software environments.

Table 7: Overview of Available Tools Across Different Domains

| Domain | Tool Count | Example Tools |
|--------|-----------|---------------|
| Chrome | 69 | `jump_to_next_tab`
`chrome_domain_data_wiper`
`open_downloads_page` |
| GIMP | 88 | `save_image_as`
`undo_last_action`
`swap_foreground_background_colors` |
| LibreOffice General | 41 | `open_find_and_replace`
`open_print_preview`
`open_hyperlink_dialog` |
| LibreOffice Calc | 114 | `spreadsheet_column_formula_injector`
`batch_spreadsheet_numeric_formatter`
`navigate_to_end_of_data_right` |
| LibreOffice Impress | 75 | `set_line_spacing_1`
`insert_non_breaking_space`
`apply_subscript` |
| LibreOffice Writer | 123 | `select_to_start_of_next_page`
`select_to_start_of_paragraph`
`apply_double_underline` |
| System | 4 | `open_system_terminal_and_execute`
`open_app_or_filename`
`switch_applications` |
| Thunderbird | 119 | `open_message_in_conversation`
`delete_message_permanently`
`search_messages_advanced` |
| VLC | 83 | `set_video_as_wallpaper`
`volume_up`
`jump_1_minute_forward` |
| VS Code | 135 | `add_vs_code_keybinding`
`vscode_exclude_folders`
`search_within_current_file` |
| **Total** | **881** | |

### A.4 DETAILS FOR SYNTHETIC TASKS

We generated a comprehensive synthetic dataset of 17,864 tasks across 10 application domains using two complementary approaches. As shown in Table 8, the evaluator-first approach contributed 4,387 high-quality tasks with complex multi-step instructions, while the instruction-first approach generated 13,477 tasks to ensure broad coverage of application functionalities.

The dataset spans diverse applications from productivity tools (LibreOffice suite with 5,885 combined tasks) to specialized software like GIMP (1,121 tasks) and development environments like VS Code (1,990 tasks). Chrome represents the largest single-domain category with 2,826 tasks, reflecting the importance of web interactions. The multi-apps category (2,113 tasks) specifically

tests cross-application workflows. Task complexity varies from simple operations (e.g., *"Change the text alignment to Center"*) to sophisticated procedures requiring multiple coordinated actions (e.g., *"Convert video to MP4 and save with a new filename"*), ensuring comprehensive evaluation of agents' GUI navigation and task execution capabilities.

Table 8: Overview of Synthetic Data Generation Across Different Domains

| Domain | Evaluator-First | Instruction-First | Example Instructions | Total |
|---|---|---|---|---|
| Chrome | 751 | 2,075 | *Find hotels in Paris for 2 adults for three nights starting next Friday and sort the list by lowest price.*

*Restore the previous session pages in Google Chrome.* | 2,826 |
| GIMP | 401 | 720 | *Please replace the current white backdrop with a solid green color, but keep the black circle in the centre exactly as it is.*

*In GIMP, navigate to the Display section and set the check style to Medium checks.* | 1,121 |
| LibreOffice Calc | 651 | 1,496 | *Open the spreadsheet and make the entire header row (row 1) bold.*

*Protect the sheet Sheet2 in LibreOffice Calc.* | 2,147 |
| LibreOffice Impress | 501 | 1,397 | *Make every slide in this deck use a solid dark-green background (RGB 0 128 0). I'd like all the pages to share that exact colour so the presentation looks consistent.*

*Add a video from /videos/video3.mov to slide 3 in LibreOffice Impress.* | 1,898 |

*Continued on next page*

Table 8 – *Continued from previous page*

| Domain | Evaluator-First | Instruction-First | Example Instructions | Total |
|---|---|---|---|---|
| LibreOffice Writer | 851 | 989 | *Change the default font in LibreOffice Writer to Calibri.*

*Change the text alignment to Center in LibreOffice Writer.* | 1,840 |
| OS/System | 301 | 1,197 | *I accidentally created a file called "draft.txt" on my Desktop. Please delete it completely so it's no longer there.*

*View the partitioning table of the disk named {disk_name} in the Disks app.* | 1,498 |
| Thunderbird | 351 | 1,084 | *Create a new folder named "ToSort" inside the Local Folders section.*

*Import contacts from Windows Mail into Thunderbird.* | 1,435 |
| VLC | 330 | 666 | *Open the cat photo in VLC and set it as my desktop wallpaper.*

*Play the current video in VLC Media Player.* | 996 |
| VS Code | 250 | 1,740 | *Could you open VS Code and create a new text file named "meeting_notes.txt" inside the folder "/home-/user/notes"? Make sure to save the file before you finish.*

*Search for the term Data Structure in the document and highlight it in LibreOffice Writer* | 1,990 |

Table 8 – *Continued from previous page*

| Domain | Evaluator-First | Instruction-First | Example Instructions | Total |
|---|---|---|---|---|
| Multi-apps | - | 2,113 | *Change the desktop wallpaper to Desert on the Ubuntu desktop.* *Search for JavaScript in Brave settings and enable it.* | 2,113 |
| Total | 4,387 | 13,477 | | 17,864 |

To ensure our synthetic training data prepares the agent for complex, real-world interactions, we further analyze the diversity of the generated tasks in terms of semantic coverage and trajectory length. Figure 5 visualizes these distributions.

**Semantic Diversity and Coverage.** As shown in Figure 5a, our synthetic pipeline achieves a high degree of semantic coverage, with 90.3% of the top-25 verb-noun pairs (677/750) represented in the dataset. Beyond these top frequent pairs, the dataset covers a broad spectrum of scenarios, encompassing a total of 1,285 unique actions and 5,347 unique objects. While common operations like "Open File" or "Save Document" are well-represented, the heatmap also reveals significant coverage of less common interactions (e.g., "Export Slide," "Filter Sheet"). This extensive variety—spanning over five thousand distinct interface objects—suggests that the generated tasks are not limited to

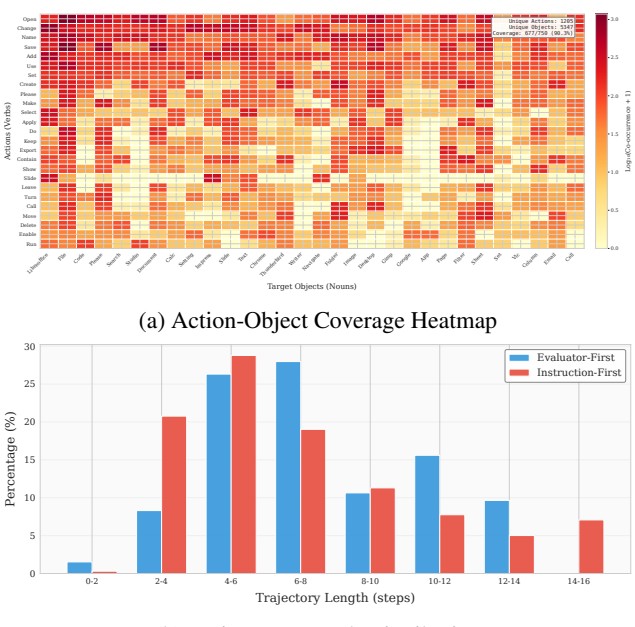

(a) Action-Object Coverage Heatmap

(b) Trajectory Length Distribution

Figure 5: **Analysis of Synthetic Task Diversity.** **(a)** Heatmap illustrating the co-occurrence frequency of the top-25 verbs and nouns in our dataset. The large volumn of actions and entities (1205 and 5347) and high density (90.3% coverage) across diverse action-object pairs confirms that our synthesis pipeline generates varied semantic scenarios rather than repetitive templates. **(b)** Distribution of trajectory lengths for Evaluator-First and Instruction-First generation strategies. The broad spread indicates that our dataset covers a wide spectrum of difficulty, from simple atomic actions to long-horizon multi-step tasks.

trivial patterns but encompass a rich diversity of desktop operating system scenarios, effectively preventing the agent from overfitting to a narrow set of instructions.

**Task Complexity and Horizon.** Figure 5b demonstrates the complexity of our synthetic tasks through the distribution of trajectory lengths. Both generation strategies—*Evaluator-First* and *Instruction-First*—produce tasks with a wide range of horizons. While a significant portion of tasks falls within the 4-8 step range, there is a substantial long-tail distribution extending up to 14-16 steps. This diversity in trajectory length is crucial for training robust agents; it ensures the model learns to maintain context and execute plans over extended periods, rather than merely solving short-horizon, immediate-reward problems.

## A.5 QUALITATIVE EXAMPLES

To illustrate the practical advantages of our hybrid control paradigm, we present three representative examples in Figures 6, 8, and 7. These cases highlight how UltraCUA strategically selects between high-level programmatic tools and low-level GUI actions to enhance efficiency, tackle complex problems, and ensure robust execution.

In the first example (Figure 6), the agent is asked to clear YouTube browsing history. Instead of relying on a potentially brittle sequence of clicks through menus, it initiates the workflow with a single programmatic tool call, `open_history_page`, to navigate directly to the correct settings page. Subsequently, it seamlessly transitions to primitive GUI actions—typing into a search field and clicking buttons—to perform the more nuanced task of filtering and deleting the specific entries. This demonstrates a practical fusion of programmatic speed for navigation and GUI flexibility for manipulation.

A more complex scenario in Figure 8 showcases the model's ability to automate workflows that are intractable for purely GUI-based agents. When tasked with batch-processing images on the desktop, UltraCUA correctly identifies the need for a scripted solution. It programmatically opens a system terminal, installs the necessary software (`imagemagick`), and proceeds to write and execute a multi-line shell script to automate the entire process. This ability to generate and utilize code represents a significant leap in problem-solving capability.

Finally, the email-starring task (Figure 7) exemplifies the agent's capacity for intelligent and fluid alternation between control modes. The process begins with a precise low-level GUI click to select the target "Bills" folder, effectively setting the context. Immediately following this, the agent switches to high-level, reliable tool calls—`select_all` and `add_or_remove_star`—to execute the core bulk operation. This strategic handoff from a specific GUI action to general-purpose tools ensures both precision and operational robustness.

## A.6 IMPACT OF FORMAT REWARD AND TOOL REWARD

In this section, we investigate the impact of two key reward shaping components: the format reward and the tool-use reward. We conduct ablation studies to justify our design choices for the final reward function.

**Format Reward.** We initially hypothesized that an explicit format reward ($R_f$) would accelerate the learning of complex action syntax. However, as shown in Figure 9(a), we observe that a strong format reward ($R_f = 0.5$) negatively impacts the final outcome reward. The results suggest that the policy becomes biased towards optimizing local syntactic correctness rather than long-term task completion.

Table 9: Effect of tool-use reward on UltraCUA-7B's performances on OSWorld.

| Configuration | SR (%) | Avg. Steps |
|---|---|---|
| $R_{\text{tool}} = 0$ | 27.9 | 9.06 |
| $R_{\text{tool}} = 0.1$ | 28.7 | 8.88 |
| $R_{\text{tool}} = 0.3$ | **28.9** | 8.81 |

Conversely, when we remove the format reward, the model achieves significantly higher outcome rewards. Although the model learns the correct format at a slightly slower pace without this explicit signal, it still converges to the correct syntax via environment feedback. Consequently, we remove the format reward in our final configuration to prioritize task success.

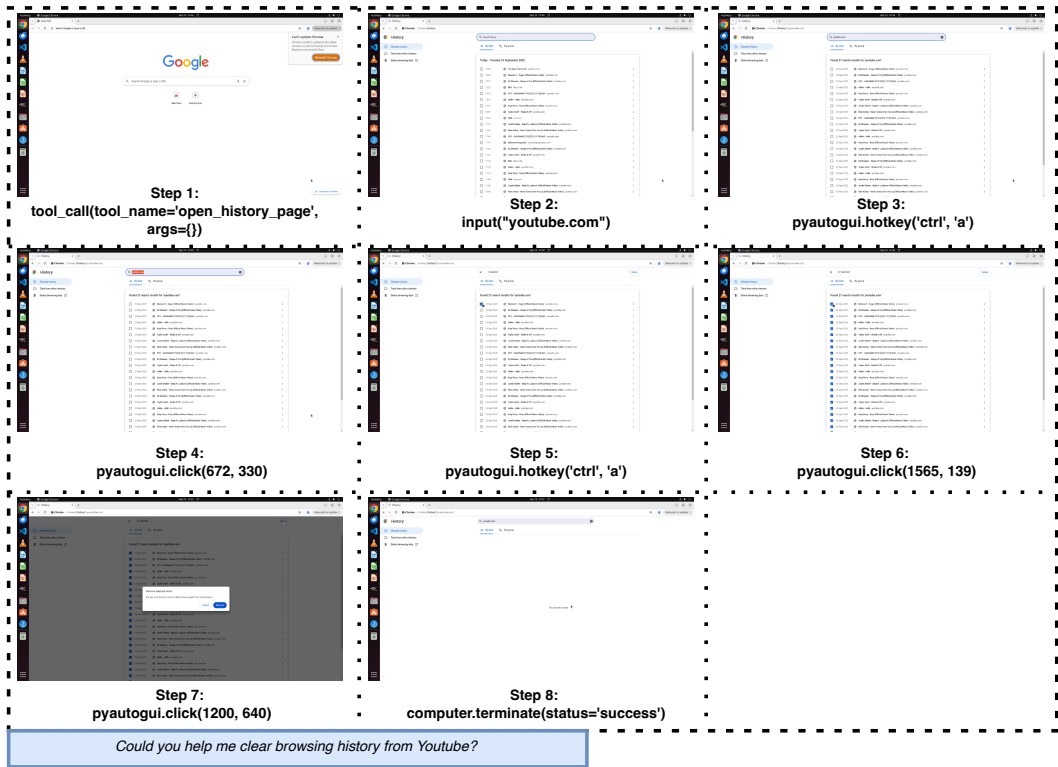

Figure 6: An example of UltraCUA-32B helping clearing certain Chrome history with hybrid control. The agent calls programmatic tool at the first step to assist directly going into the desired page.

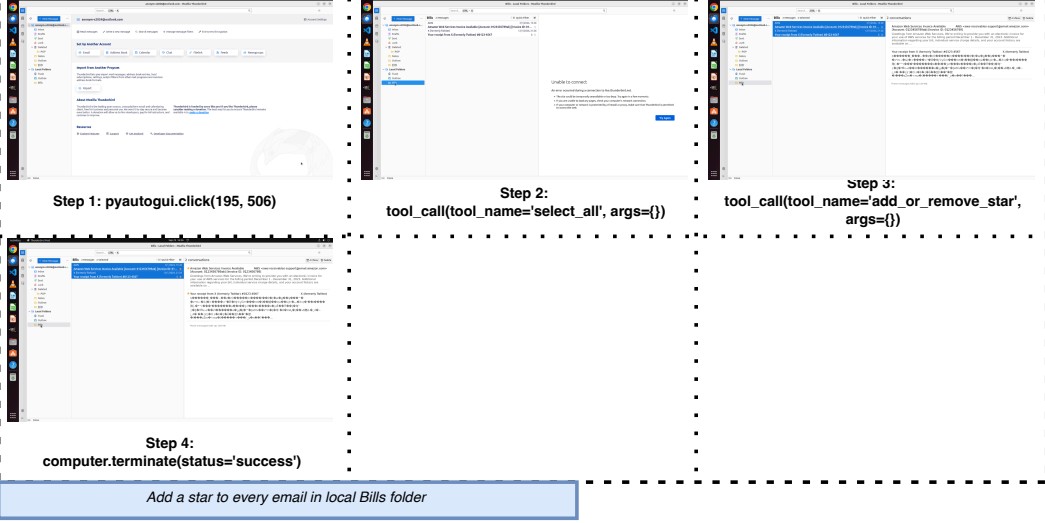

Figure 7: An example of UltraCUA-32B helping processing emails. The agent alternates between low-level actions and programmatic tool calls smartly, leading to efficient completion of the task.

**Tool-use Reward.** We further analyze the effect of the tool-use reward ($R_{\text{tool}}$), which provides a dense signal to encourage the usage of external tools. Figure 9(b) demonstrates that including a tool-use reward consistently improves the outcome reward throughout the training process compared to the baseline ($R_{\text{tool}} = 0$). Table 9 provides a quantitative comparison, where increasing $R_{\text{tool}}$ from 0 to 0.3 improves the success rate (SR) from 27.9% to 28.9% and reduces the average steps required.

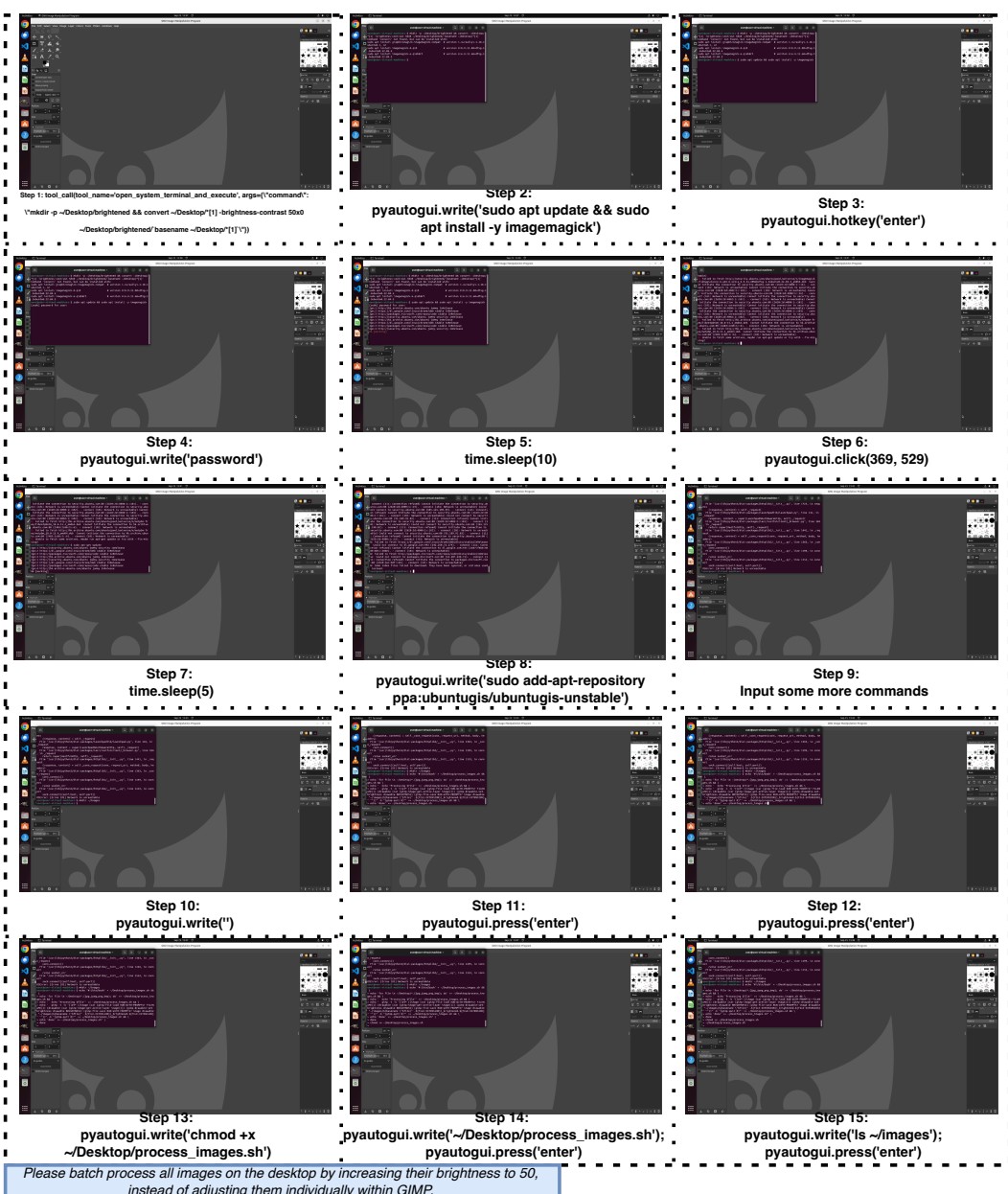

Figure 8: An example of UltraCUA-32B helping processing images with hybrid control. The model starts coding at the very first step by calling the terminal tool, and finally wrote a bash script and executed it to make the task successful.

Notably, the performance gain is also observable at $R_{\text{tool}} = 0.1$ (28.7% SR), indicating that our method is relatively robust to this hyperparameter as long as it is maintained within a reasonable range. Based on these results, we adopt $R_{\text{tool}} = 0.3$ for our main experiments to maximize agent efficiency and success.

## A.7 ERROR ANALYSIS AND COMPARISON

To understand the failure modes of our approach compared to standard GUI agents, we conducted a fine-grained error analysis on the OSWorld test tasks. We employed GPT-5 as an automated evaluator, providing it with the full multimodal trajectories—including both textual logs and screen-

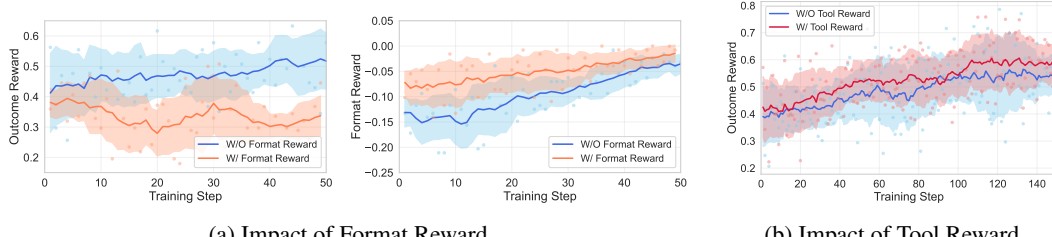

(a) Impact of Format Reward            (b) Impact of Tool Reward

Figure 9: **Ablation analysis of reward shaping. (a) Impact of Format Reward** ($R_f = 0.5$)**:** Imposing a strong format reward biases the policy towards optimizing complex syntax, detrimentally affecting the final outcome reward. Removing the explicit format reward allows the model to achieve superior outcome performance while still gradually learning the correct format. **(b) Impact of Tool-use Reward** ($R_{tool} = 0.3$): Including an auxiliary tool-use reward acts as a beneficial dense signal, resulting in consistently higher outcome rewards compared to the baseline.

shots—to diagnose the primary cause of each failure. The error distribution is visualized in Figure 10.

**Reduced Perception and Memory Errors.** Compared to the GUI-only baseline (UI-TARS-1.5-7B), UltraCUA-7B demonstrates a significant reduction in both *GUI Misunderstandings* (450 vs. 479) and *State Management Failures* (223 vs. 365). We attribute the reduced GUI errors to our specialized training strategy, which fosters a more robust representation of user interface elements. Furthermore, the substantial drop in state management failures highlights the effectiveness of our working memory mechanism, which enables the agent to maintain context over long-horizon tasks more effectively than the baseline.

**Challenges in Hybrid Control.** Conversely, UltraCUA-7B exhibits a higher incidence of *Action/Tool Malfunctions* (15.6% vs. 10.5%). This category encompasses two distinct failure types: (1) semantically incorrect tool selection given the current state, and (2) syntactic

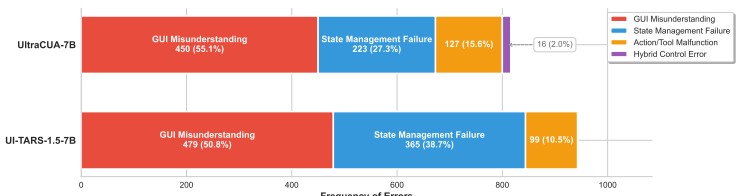

Figure 10: **Error distribution analysis on OSWorld test tasks.** We compare the frequency of failure modes between UltraCUA-7B (UltraCUA) and the GUI-only baseline (UI-TARS-1.5-7B).

errors during tool invocation. The increase in this error category is an expected trade-off of the hybrid control framework; unlike the baseline which relies solely on standardized GUI interactions, UltraCUA-7B must acquire and manage a diverse set of external tools with complex and varying syntactic requirements, increasing the difficulty of precise action execution.

**Qualitative Case Studies.** To substantiate these quantitative findings, Figure 11 presents a qualitative analysis of real-world failure cases. These examples illustrate specific behavioral patterns, such as the grounding failures in *GUI Misunderstanding* where the agent clicks non-interactive regions, and the strategic conflicts observed in *Hybrid Control Errors*. In the latter, as shown in the bottom-right case, the agent correctly identifies the goal but selects a suboptimal modality—needlessly invoking a tool when a direct GUI interaction (double-click) would have been more efficient. This highlights the non-trivial challenge of learning an optimal policy for modality switching.

## A.8 DETAILS FOR THE RL TRAINING AND HYBRID CONTROL MECHANISMS

In this section, we formalize the core algorithmic components of our framework. Algorithm 1 details the reinforcement learning update procedure, which utilizes the DAPO objective to optimize the policy based on binary success signals from the computer-use environment. Algorithm 2 outlines the hybrid control loop employed during inference and training rollouts, demonstrating how the agent dynamically switches between low-level GUI operations ($\mathcal{A}_{GUI}$) and high-level programmatic tool invocations ($\mathcal{T}$) to solve tasks efficiently.

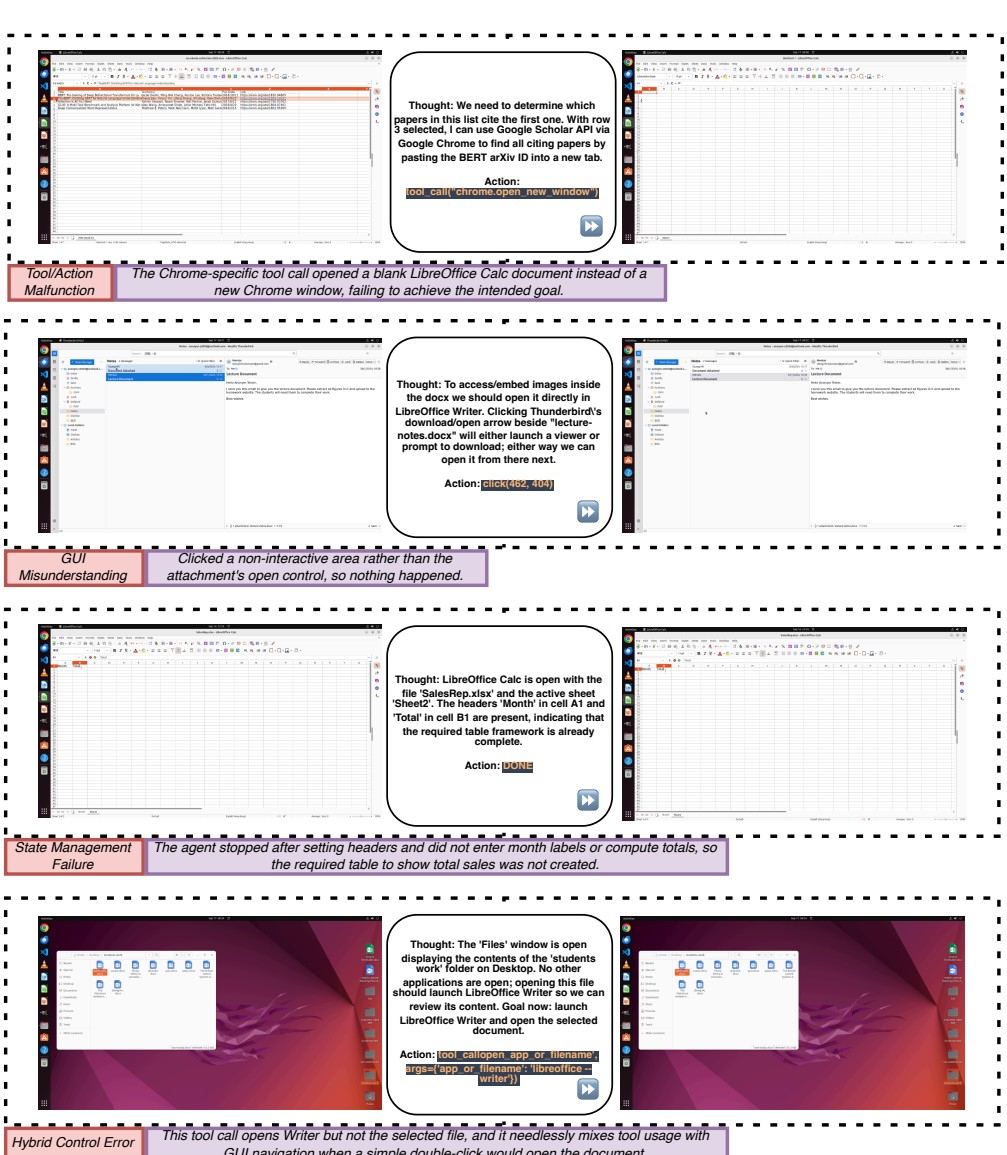

Figure 11: Real-world cases for the four common error types of UltraCUA.

**Algorithm 1** The RL update procedure

1: **Input:** Dataset $\mathcal{D}$, Computer-use Environment $\mathcal{E}$, Policy $\pi_\theta$, Reference Policy $\pi_{\theta_{\text{old}}}$, Group size $G$, Batch size $N$, Learning rate $\eta$, Clip thresholds $\varepsilon_{\text{low}}, \varepsilon_{\text{high}}$.
2: **Initialize:** $\theta \leftarrow \theta_{\text{old}}$
3: **for** each training step **do**
4:     Sample a batch of $N$ tasks $q_{1\ldots N} \sim \mathcal{D}$
5:     **for** each task $q_j$ in batch **do**
6:         *// Policy interacts with the environment to generate trajectories*
7:         Generate $G$ trajectories $\{o_{j,i}\}_{i=1}^{G}$ via $\pi_{\theta_{\text{old}}}$ interacting with $\mathcal{E}$ initialized with $q_j$
8:         *// Get binary success/failure signal directly from the environment*
9:         Obtain correctness signal $c_{j,i}$ from $\mathcal{E}$ for each trajectory $o_{j,i}$
10:        *// Filter out samples with zero variance (all correct or all wrong)*
11:        **if** $\sum_{i=1}^{G} c_{j,i} = 0$ **or** $\sum_{i=1}^{G} c_{j,i} = G$ **then**
12:            Continue
13:        **end if**
14:        Compute rewards $\{R_{j,i}\}_{i=1}^{G}$ based on $c_{j,i}$
15:        Compute group statistics:
16:            $\mu_j = \text{mean}(\{R_{j,i}\}_{i=1}^{G}), \quad \sigma_j = \text{std}(\{R_{j,i}\}_{i=1}^{G})$
17:        **for** each trajectory $o_{j,i}$ and token $t$ **do**
18:            Compute Advantage: $\hat{A}_{j,i,t} = \frac{R_{j,i}-\mu_j}{\sigma_j}$
19:        **end for**
20:    **end for**
21:    **Update Policy:**
22:    Compute loss gradient $\nabla_\theta \mathcal{J}(\theta)$ maximizing the DAPO objective:
23:        $r_{i,t}(\theta) = \frac{\pi_\theta(o_{i,t}|q,o_{i,<t})}{\pi_{\theta_{\text{old}}}(o_{i,t}|q,o_{i,<t})}$
24:        $\mathcal{J} = \mathbb{E}\left[\frac{1}{\sum|o_i|}\sum_{i,t}\min\left(r_{i,t}(\theta)\hat{A}_{i,t}, \text{clip}(r_{i,t}(\theta), 1-\varepsilon_{\text{low}}, 1+\varepsilon_{\text{high}})\hat{A}_{i,t}\right)\right]$
25:    $\theta \leftarrow \theta + \eta\nabla_\theta\mathcal{J}(\theta)$
26:    $\theta_{\text{old}} \leftarrow \theta$
27: **end for**

**Algorithm 2** Hybrid Control Loop for UltraCUA

1: **Input:** Initial Observation $s_0$, Max Steps $N_{\max}$.
2: **Definitions:**
3:     $\mathcal{A}_{\text{GUI}}$: Set of low-level GUI actions (e.g., `click`, `type`, `scroll`).
4:     $\mathcal{T}$: Set of high-level programmatic tools $\{T_1, T_2, \ldots, T_k\}$.
5:     $a_{\text{API}}$: Special action token for invoking programmatic tools.
6: **Initialize:** Trajectory $\tau \leftarrow \emptyset$, $t \leftarrow 0$, Done $\leftarrow$ False.
7: **while** $t < N_{\max}$ **and not** Done **do**
8:     *// Agent perceives current screen and history to generate action*
9:     $a_t, \text{params}_t \leftarrow \pi_\theta(s_t, \tau)$
10:     *// Branching logic for Hybrid Control*
11:     **if** $a_t \in \mathcal{A}_{\text{GUI}}$ **then**
12:         *// Case 1: Execute Low-level GUI Interaction*
13:         Execute GUI action $a_t$ with $\text{params}_t$ on the environment.
14:         $s_{t+1}, r_t, \text{Done} \leftarrow \text{Env.step}(a_t, \text{params}_t)$
15:     **else if** $a_t = a_{\text{API}}$ **then**
16:         *// Case 2: Execute High-level Tool Call*
17:         Parse tool name $T_i$ and arguments $\theta_i$ from $\text{params}_t$.
18:         **if** $T_i \in \mathcal{T}$ **then**
19:             Execute tool function: result $\leftarrow T_i(\theta_i)$
20:             $s_{t+1}, r_t, \text{Done} \leftarrow \text{Env.update(result)}$
21:         **else**
22:             Handle invalid tool error.
23:         **end if**
24:     **else**
25:         Handle invalid action.
26:     **end if**
27:     Update trajectory: $\tau \leftarrow \tau \cup \{(s_t, a_t, \text{params}_t, r_t, s_{t+1})\}$
28:     $t \leftarrow t + 1$
29: **end while**
30: **return** Trajectory $\tau$

