# OpenReview forum: "UltraCUA: Scaling Computer Use Agent through GUI and Programmatic Control"
_ICLR.cc/2026/Conference — ICLR 2026 Conference Desk Rejected Submission_

### Official Review · Reviewer_yDXL · 2025-10-29

**Soundness:** 3
**Presentation:** 3
**Contribution:** 2
**Rating:** 2
**Confidence:** 5

**Summary:**

This paper presents UltraCUA, a scalable computer-use agent that unifies low-level GUI actions with high-level programmatic tool invocation via a hybrid control mechanism. To enable this capability, the authors develop an automated pipeline for extracting and expanding programmatic tools from software documentation, open-source code, and agent-generated scripts. Additionally, they introduce a large-scale synthetic data engine that provides more than 17,000 verifiable tasks across diverse real-world computing scenarios. UltraCUA is trained using a two-stage process, supervised fine-tuning followed by online reinforcement learning, which substantially improves tool-use strategies and execution efficiency. Experiments demonstrate strong performance gains on OSWorld benchmarks and significant cross-platform generalization to Windows environments without domain-specific tuning. Overall, this work provides a systematic and scalable paradigm for bridging primitive GUI interactions and programmatic intelligence, advancing the capability of computer-use agents toward robust and efficient automation.

**Strengths:**

- The hybrid setup provides clear usability benefits over purely GUI-driven agents, especially in reducing long action chains and improving execution efficiency.
- The automated pipeline for constructing verifiable hybrid-control tasks (17K+) is well executed and helps address the data bottleneck in computer-use agent training.
- The paper includes strong empirical evaluation across multiple environments, showing consistent improvements and demonstrating that the system works reliably in practice.

**Weaknesses:**

- The paper argues that GUI actions are low-level while programmatic tools are high-level, yet it does not clearly justify why both must coexist within a unified control framework. Prior programmatic agents (e.g., Voyager, CodeActAgent) already show that LLMs can synthesize execution scripts that effectively serve as high-level action sequences for computer manipulation. This raises the question of whether hybrid control offers conceptual necessity beyond what script-based control can already achieve.
- The RL objective adds a positive bonus for trajectories that contain tool calls (R_tool = 0.3 when succeed), which intrinsically biases the learned policy toward tools. This makes it hard to attribute the reported gains to an emergent hybrid strategy rather than to reward shaping that prefers tool usage. The paper also removes format rewards, which weakens constraints on syntactic correctness and may cause brittle tool calling.
- The paper describes an automated pipeline to harvest tools, but provides no quantitative coverage (how many APIs/tools per app), no failure/precision metrics, and no robustness to software updates—so the claimed scalability is not empirically substantiated. (Contribution described in abstract; missing robustness analyses elsewhere.)
- The paper cites both “17,000+” verifiable tasks and “16,000+” verified tasks in different sections, with limited breakdowns of task types or difficulty alignment with claimed hybrid benefits. This inconsistency complicates reproducibility and makes it hard to assess how much of the gain is due to data scale vs. hybrid design.

**Questions:**

- Given that OSWorld tasks are deterministic with rule-based evaluators and the Windows evaluation is treated as OOD primarily due to an OS change, can you evaluate on stochastic or previously unseen UIs and quantify UI/layout variability to substantiate real-world generalization claims?
- On WindowsAgentArena the absolute SR is 21.7% and results are reported at 15-step budgets; could you report wall-clock time, crash/timeout rates, and performance under larger step budgets to demonstrate end-to-end practicality, and clarify how your chosen metrics (SR/Pass@4/steps) relate to real latency and throughput?
- Since trajectory collection uses a powerful planner (o3) and a SOTA grounder (GTA1-7B), can you disentangle how much improvement comes from these components versus the proposed hybrid formulation (e.g., by swapping/removing them)?

---

> ### Author Response · Authors · 2025-11-25
> **Response to Reviewer yDXL (1/3)**
>
> We thank Reviewer yDXL for the comprehensive evaluation, particularly for recognizing the execution of our automated pipeline and the breadth of our empirical results. We value the constructive critique regarding the necessity of hybrid control and the RL formulation, which has prompted us to provide deeper clarifications and new ablation results below.
>
> ### **1. The Necessity of Hybrid Control (Addressing W1)**
>
> The reviewer questions the necessity of hybrid control given existing code-generation agents. We posit that **GUI and Programmatic controls are complementary**, and a unified framework is required to build a robust **General Computer Agent**:
>
> * **Complementarity & Accessibility:** Purely programmatic agents hit a "hard wall" in applications lacking APIs (e.g., creative software like GIMP, legacy apps). Conversely, pure GUI agents are notoriously inefficient at high-repetition tasks. Hybrid control ensures **Universal Accessibility** via GUI primitives while leveraging **Programmatic Efficiency** for complex logic.
> * **Strategic Alternation (Qualitative Evidence):** Our qualitative analysis (Appendix A.5) demonstrates this synergy in practice:
>     * **Navigation efficiency (Figure 6):** In a Chrome task, the agent uses a tool (`open_history_page`) to instantly skip deep menu navigation, then transitions to GUI primitives for the specific interaction of deleting entries.
>     * **Solving "Intractable" GUI Tasks (Figure 8):** For batch image processing—a task intractable for pure GUI agents due to repetition—UltraCUA switches to the terminal to write and execute a shell script (`imagemagick`), solving the problem programmatically.
>     * **Contextual Handoff (Figure 7):** In email management, the agent uses a precise GUI click to select a specific folder ("Bills"), then immediately hands off to high-level tools (`select_all`, `add_or_remove_star`) to robustly execute the bulk operation.
>
> These examples illustrate that hybrid control is not merely a combination of existing methods, but a **conceptual necessity** for general automation. It enables the agent to dynamically select the optimal modality—GUI for visual precision and generalization, or Code for logic and speed—transcending the limitations inherent to either approach in isolation.
>
> ### **2. Reward Shaping and Tool Bias (Addressing W2)**
>
> We address the concern that the tool-use reward ($R_{tool}=0.3$) might artificially bias the policy. We argue that this dense signal is actually necessary to **prevent** the agent from converging to a local optimum of "pure GUI" interaction.
>
> * **Counteracting Risk Aversion:** In the early stages of training, tool invocation is "risky" due to complex parameter requirements, whereas GUI clicking is "safe" but inefficient. Without $R_{tool}$, the agent tends to revert to inefficient GUI actions to avoid the penalty of failed tool calls, effectively "hacking" the system by ignoring the hybrid capabilities. The $R_{tool}$ signal encourages the agent to overcome this initial barrier and master the more efficient hybrid control.
> * **Format Reward Paradox:** We also address the reviewer's concern regarding the removal of format rewards. As illustrated in our new **Figure 9 (Appendix A.6)**, imposing a strong format reward ($R_f$) ironically leads to reward hacking: the agent learns to output syntactically correct but semantically useless tool calls just to accumulate format points. By removing $R_f$, we force the agent to value syntax *only* insofar as it contributes to the final task success (Outcome Reward), aligning the incentive structure with the true objective.
>
> **Table 1: Tool-Use Reward Sensitivity Analysis (Appendix A.6)**
>
> | $R_{\text{tool}}$ | Success Rate | Avg. Steps |
> | :--- | :--- | :--- |
> | 0.0 | 27.9% | 9.06 |
> | 0.1 | 28.7% | 8.88 |
> | 0.3 (Ours) | **28.9%** | **8.81** |
>
> **Empirical Choice:** As shown above, $R_{tool}=0.3$ was empirically selected as it yields the highest success rate with the fewest steps. We further direct the reviewer to **Figure 9**, which visually demonstrates that $R_{tool}=0.3$ consistently lifts the outcome reward curve compared to the baseline, while removing $R_f$ prevents the "syntax-over-substance" optimization trap.

---

> > ### Author Response · Authors · 2025-11-25
> > **Response to Reviewer yDXL (2/3)**
> >
> > ### **3. Quantitative Analysis of Tool Pipeline (Addressing W3)**
> >
> > We appreciate the reviewer's request for more granular metrics regarding our tool collection pipeline. We address this in three parts: existing coverage, scalability verification, and specific precision metrics.
> >
> > **1. Existing Coverage & Demonstration:**
> > We respectfully direct the reviewer to **Table 7** in the submission, where we provided a detailed quantitative breakdown of the tool library and listed specific tool examples for each application. These definitions cover the core functionality of the target applications.
> >
> > **2. Scalability Verification:**
> > To empirically substantiate our scalability claims, we refer to **Section 3.4.1 (Table 5)**. Here, we conducted tool scaling experiments, demonstrating that as we increase the number of available tools during test-time, the model effectively learns to acquire relevant tools without performance degradation.
> >
> > **3. Precision Metrics & Validation Strategy:**
> > We adopt a tiered validation strategy based on the tool source:
> > * **Documentation & Open Source:** We treat tools harvested from official software documentation and verified open-source repositories as **valid ground truth**. These sources are maintained by human developers and reflect the official API specifications, ensuring high reliability by default.
> > * **Agent-Generated Tools:** For tools synthesized by our Coding Agent, we implement a rigorous validation pipeline involving execution and unit testing. We provide the specific precision metrics for this component below:
> >
> > **Table 2: Coding Agent Pipeline Validation Statistics**
> >
> > | Metric | Count |
> > | :--- | :--- |
> > | Total Runs on Synthetic Tasks | 241 |
> > | Trajectories attempting `coding-as-action` | 151 |
> > | Successful Trajectories (Task Solved) | 39 |
> > | **Extracted Potential Tools** | **58** |
> > | **Valid Tools (Passed Generated Unit Tests)** | **56** |
> > | **Validity Rate** | **96.5%** |
> >
> > Only 2 out of 58 extracted tools failed the subsequent agentic unit tests, demonstrating high precision in our extraction pipeline.
> >
> > **4. Robustness to Software Updates:**
> > We address the concern regarding software version changes:
> > * **Documentation-based Tools:** These are highly robust. The parsing pipeline is automated and schedulable; updating the tool library requires only re-running the scraper on the latest documentation, ensuring zero lag between software updates and agent capabilities.
> > * **Code-based Tools:** We acknowledge that scripts can be brittle if underlying APIs change. However, our Foundation Model approach mitigates this better than static libraries. If a tool fails due to an update, the model can detect the execution error and, using its coding capability, generate an updated tool definition in-context. This allows for rapid adaptation without the need for expensive weight retraining.
> >
> > ### **4. Data Consistency and Task Analysis (Addressing W4)**
> >
> > We apologize for the typo regarding task counts; we have corrected the manuscript to reflect the accurate count of **17,000+ verifiable tasks**.
> >
> > Regarding task breakdowns, we respectfully refer the reviewer to **Table 8** in the submission, which details the distribution across domains. To further substantiate the diversity and difficulty alignment, we have added **Figure 5 (Appendix A.3)**:
> >
> > * **Semantic Diversity (Fig 5a):** A heatmap of top-25 verbs/nouns reveals a vast action space (1,205 actions, 5,347 entities) with 90.3% coverage across diverse action-object pairs. This confirms that our synthesis pipeline generates varied semantic scenarios rather than repetitive templates.
> > * **Difficulty Spectrum (Fig 5b):** The distribution of trajectory lengths shows a broad spread, indicating that our dataset covers a wide spectrum of difficulty—from simple atomic actions to long-horizon multi-step workflows. This diversity is essential for training the hybrid policy to handle both simple GUI tasks and complex, tool-heavy problems.

---

> > > ### Author Response · Authors · 2025-11-25
> > > **Response to Reviewer yDXL (3/3)**
> > >
> > > ### **5. Generalization across Distinct Environments (Addressing Q1)**
> > >
> > > We respectfully clarify that the evaluation on Windows constitutes a rigorous **Out-of-Distribution (OOD) test**. It is not merely a change in OS theme; the agent faces a disjoint set of applications and a fundamentally different interaction paradigm that was **absent** from the training set:
> > >
> > > | Domain | Ubuntu (Training) | Windows (Evaluation) | **Key OOD Factor** |
> > > | :--- | :--- | :--- | :--- |
> > > | **System Control** | Nautilus, GNOME Settings, Bash Terminal | **File Explorer**, Windows Settings, PowerShell | **Disjoint Logic:** Different file paths and navigation flows. |
> > > | **Exclusive Apps** | **GIMP**, **Thunderbird** | **Paint** (Simple Editing), **Notepad**, **Clock** | **Unseen Software:** The agent must transfer concepts to entirely new interfaces. |
> > > | **Visual Layout** | GTK / GNOME Menus | WinUI | **Paradigm Shift:** WinUI is unseen in Ubuntu. |
> > >
> > > **Implications for Generalization:**
> > > The agent must generalize from Linux menus to Windows Ribbons. Notably, **~24% of tasks** rely on software *never seen* during training. Achieving a **21.7% Success Rate** here demonstrates robust, generalized control policies rather than pixel memorization.
> > >
> > > ### **6. Computational Efficiency and End-to-End Practicality (Addressing Q2)**
> > >
> > > We provide the requested breakdown of wall-clock time and performance under extended step budgets.
> > >
> > > **Infrastructure Note:**
> > > Absolute wall-clock times differ between environments due to hosting infrastructure:
> > > * **OSWorld:** Hosted on remote clusters using **nested virtualization** (higher base latency).
> > > * **WindowsAgentArena:** Hosted on local **bare-metal** machines (lower base latency).
> > > * **Inference:** All models utilize the **vLLM** asynchronous inference engine.
> > >
> > > **Table 4: Performance vs. Time Budget**
> > >
> > > | Model | Domain | 15-Step SR | 15-Step Time (s) | 50-Step SR | 50-Step Time (s) |
> > > | :--- | :--- | :--- | :--- | :--- | :--- |
> > > | UI-TARS-1.5-7B | OSWorld | 23.4% | 284.9 | 26.1% | 452.7 |
> > > | | Windows | 18.1% | 71.6 | 26.1% | 86.3 |
> > > | **UltraCUA-7B** | **OSWorld** | **28.9%** | **273.4** | **30.2%** | **490.9** |
> > > | | **Windows** | **21.7%** | **73.4** | **28.2%** | **90.0** |
> > >
> > > **Findings & Clarifications:**
> > >
> > > * **Metric Selection (Steps vs. Time):** We posit that **Step Count** is a more representative metric for agent efficiency than wall-clock time, as it is agnostic to the underlying hardware, hosting method (virtualized vs. bare-metal), or inference engine optimization.
> > > * **Relation to Real Latency:** The data confirms that **reducing steps directly improves throughput**. Although UltraCUA requires a larger context window to encode tool definitions (adding slight overhead per generation), it solves tasks in fewer steps by replacing lengthy GUI interaction chains with efficient programmatic calls. This efficiency gain outweighs the context overhead, resulting in **on-par or lower wall-clock time** (e.g., 273.4s vs 284.9s on OSWorld).
> > > * **Stability:** Regarding crash/timeout rates, we observed negligible instability (<1%) across all evaluations. We consider stability primarily a function of the serving infrastructure (handled here by vLLM) rather than a limitation of the agent architecture itself.
> > >
> > > ### **7. Disentangling Component Contributions (Addressing Q3)**
> > >
> > > The reviewer correctly identifies that our data generation pipeline leverages powerful components (o3 planner and GTA-1 grounder). To isolate the specific contribution of the **Hybrid Control formulation**, we conducted a controlled comparison where the data source was held constant while only the control architecture varied (refer to Section 3.3, Table 3).
> > >
> > > * **Control (Pure GUI SFT):** A model trained on high-quality trajectories generated by o3 + GTA, but restricted to only GUI primitives. **Success Rate: 25.1%**
> > > * **Experimental (Hybrid SFT):** A model trained on the *same distribution* of o3 + GTA generated tasks, but enabled with the hybrid control space. **Success Rate: 27.0%**
> > >
> > > **Conclusion:**
> > > Since both models distilled knowledge from the exact same components (o3/GTA), the performance gain (+1.9% absolute at the SFT stage) is strictly attributable to the **superior expressivity and efficiency of the Hybrid Control architecture**, rather than the quality of the planner or grounder alone. This confirms that even with SOTA supervision, a pure GUI policy hits a performance ceiling that the hybrid formulation can transcend.
> > >
> > > ***
> > >
> > > We deeply appreciate your rigorous review, which has pushed us to significantly strengthen our empirical analysis and clarify the core motivations behind our design. We hope the new experiments and detailed breakdowns have resolved your concerns regarding the necessity and validity of the hybrid approach. We remain fully available to discuss these points further or address any additional questions you may have during the rebuttal period.

---

### Official Review · Reviewer_jRhg · 2025-10-31

**Soundness:** 3
**Presentation:** 3
**Contribution:** 2
**Rating:** 6
**Confidence:** 3

**Summary:**

The paper presents UltraCUA, a novel approach to enhancing computer-use agents by integrating graphical GUI actions with high-level programmatic tool calls. This hybrid control model addresses the limitations of traditional CUAs that rely solely on primitive actions like clicking and typing, which often lead to cascading errors and inefficiencies. UltraCUA introduces four key components: an automated tool collection pipeline, a synthetic data engine for task generation, a hybrid control trajectory collection, and a two-stage training pipeline combining supervised fine-tuning with reinforcement learning. The experiments demonstrate significant improvements over state-of-the-art models, with UltraCUA achieving a 27% relative improvement in success rate on the OSWorld benchmark and a 21.7% success rate in out-of-domain evaluations on WindowsAgentArena.

**Strengths:**

1. The integration of GUI actions with programmatic tool calls is a significant advancement, allowing for more efficient and reliable task execution. This approach effectively bridges the gap between GUI-based generality and programmatic efficiency.

2. The paper outlines a robust methodology, including an automated tool collection pipeline and a dual-pipeline synthetic data engine, which ensures the generation of diverse and verifiable tasks. This comprehensive approach supports the development of a scalable and adaptable agent.

3. UltraCUA demonstrates substantial improvements over existing models, both in terms of success rates and execution efficiency. The model's ability to generalize across different platforms, as evidenced by its performance on WindowsAgentArena, highlights its versatility and robustness.

**Weaknesses:**

1. The manuscript does not adequately discuss the limitations of the proposed approach. A thorough examination of potential drawbacks or areas for improvement would provide a more balanced view and help guide future research

2. The paper does not clearly differentiate its contributions from previous works, which may lead readers to perceive it as merely a combination of existing methods rather than a novel advancement.

3. The technique contribution could be RL, but which is a little bit simple of the hyper-parameters in the reward setting, which could give a further ablation study and discussion.
There is a concern that the reward system could be hacked by using tools that do not affect the environment but still contribute to the reward.

4. The necessity of hybrid control should be further discussed, e.g., what if using a router model to decide between GUI primitive actions and programmatic tool calls?

5. There are some typos, like line 126 ”tool”.

**Questions:**

see the weakness.

---

> ### Author Response · Authors · 2025-11-25
> **Response to Reviewer jRhg (1/2)**
>
> We thank Reviewer jRhg for the detailed assessment and for recognizing UltraCUA as a "significant advancement" that "effectively bridges the gap between GUI-based generality and programmatic efficiency." We appreciate the constructive feedback regarding the discussion of limitations and the RL formulation. We have updated our manuscript to address these points and provide detailed responses below.
>
> ### **1. Discussion of Limitations and Failure Analysis (Addressing W1)**
>
> We agree that a detailed discussion of limitations was necessary. We have added a **comprehensive failure analysis in Appendix C (visualized in the new Figure 10)**, using GPT-5 to classify failure modes on the same subset of OSWorld failed runs for both models.
>
> **Table 1: Quantitative Error Distribution (Appendix C)**
>
> | Error Type | UltraCUA-7B | UI-TARS-1.5-7B | Absolute Change |
> | :--- | :--- | :--- | :--- |
> | **GUI Misunderstanding** | **450** | 479 | -29 |
> | **State Management Failure** | **223** | 365 | **-142** |
> | **Action/Tool Malfunction** | **127** | 99 | +28 |
> | **Hybrid Control Error** | **16** | 0 | +16 |
>
> **Key Insights & Limitation Discussion:**
> * **Complexity Trade-off:** The hybrid framework trades a slight increase in tool-related complexity (+28 "Action/Tool Malfunction" cases) for a massive reduction in "State Management Failure" (-142 cases). Programmatic tools allow the agent to offload memory-intensive logic, preventing the cascading context loss typical in long-horizon GUI tasks.
> * **Alleviating Visual Dependency:** We observe a notable reduction in "GUI Misunderstanding" errors (-29 cases). This supports our hypothesis that hybrid control mitigates visual grounding bottlenecks: when facing ambiguous or cluttered UIs, the agent can strategically bypass visual hazards by invoking programmatic tools (e.g., using `search_file` instead of visually scanning a directory), thereby reducing the probability of visual grounding errors.
>
> ### **2. Differentiation and Contributions (Addressing W2)**
>
> We appreciate the reviewer pointing out the need for sharper differentiation. UltraCUA is the **first multi-modal foundation model to demonstrate unified GUI + API hybrid control specifically for general computer use**. Our contributions beyond prior art include:
>
> 1.  **Automated Tool Discovery Pipeline** (from documentation and code).
> 2.  **Verifiable Synthetic Data Engine** (17,000+ hybrid tasks).
> 3.  **Hybrid Trajectory Generation System**.
> 4.  **Dual-Stage Hybrid Training Strategy** (SFT + Online RL).
>
> **Commitment to Revision:**
> We commit to revising the **Introduction** and **Related Work** sections in the future version to explicitly highlight these unique contributions and contrast them with prior art.
>
> ### **3. RL Ablation and Reward Hacking Concerns (Addressing W3)**
>
> We appreciate the scrutiny on the RL setup. We intentionally kept the formulation streamlined to ensure stability, but we agree that hyperparameter sensitivity and potential reward hacking are valid concerns.
>
> **Response to "Reward Hacking":**
> We argue that our specific reward design actually **prevents** the hacking behaviors the reviewer is concerned about:
>
> * **Counteracting Risk Aversion (Why $R_{tool}$ is needed):** In early training, tool invocation is "risky" due to complex syntax requirements, whereas GUI clicking is "safe" but inefficient. Without $R_{tool}$, the agent tends to revert to inefficient GUI actions to avoid the penalty of failed tool calls—effectively "hacking" the system by ignoring its hybrid capabilities. The $R_{tool}$ signal serves as a necessary scaffold to encourage the agent to overcome this initial barrier and master the more efficient hybrid control.
> * **The Format Reward Paradox:** To directly address the concern that "tools that do not affect the environment still contribute to reward," we highlight our decision to **remove Format Rewards ($R_f$)**. As illustrated in our new **Figure 9 (Appendix A.6)**, imposing a strong format reward ($R_f$) ironically leads to true reward hacking: the agent learns to output syntactically correct but semantically useless tool calls just to accumulate format points. By removing $R_f$, we force the agent to value syntax *only* insofar as it contributes to the final task success (Outcome Reward).
>
> **Table 2: Tool-Use Reward Sensitivity Analysis (Appendix A.6)**
>
> | $R_{\text{tool}}$ | Success Rate | Avg. Steps |
> | :--- | :--- | :--- |
> | 0.0 | 27.9% | 9.06 |
> | 0.1 | 28.7% | 8.88 |
> | **0.3 (Ours)** | **28.9%** | **8.81** |
>
> **Empirical Verification:**
> As shown above, $R_{tool}=0.3$ was empirically selected as it yields the highest success rate with the fewest steps. We further direct the reviewer to **Figure 9**, which visually demonstrates that this setting consistently lifts the outcome reward curve compared to the baseline, confirming that the agent is optimizing for task completion rather than exploiting the reward function.

---

> > ### Author Response · Authors · 2025-11-25
> > **Response to Reviewer jRhg (2/2)**
> >
> > ### **4. Necessity of Hybrid Control vs. Router (Addressing W4)**
> >
> > The reviewer raises an excellent point regarding the use of a modular router. We argue that a **Unified Foundation Model** is superior to a Router-based architecture for Computer Use Agents for two key reasons:
> >
> > 1.  **Latency and Context Fragility:** A router introduces significant latency (double inference) and context management overhead. Transferring full multimodal context (high-res screenshots and history) between modules is computationally expensive. Compressing this context for a router often leads to information loss, causing suboptimal decisions (e.g., missing a visual error state that dictates a modality switch), thereby introducing structural risks to task completion that a unified model avoids.
> > 2.  **End-to-End Optimization:** A router creates a non-differentiable bottleneck. By training a single model via RL, UltraCUA learns the *strategic* value of switching modalities based on expected long-term reward. This avoids the complexity of maintaining separate engines and allows the policy to optimize modality switching jointly with action execution.
> >
> > ### **5. Typos**
> >
> > We thank the reviewer for catching the typo on line 126. We have corrected it.
> >
> > ***
> >
> > We have endeavored to address the weaknesses you identified through these detailed clarifications and the new quantitative analyses in the Appendix. We respectfully hope these updates strengthen your assessment of our contribution. We would be grateful for the opportunity to engage in further discussion should you have any additional questions or concerns during the rebuttal period.

---

### Official Review · Reviewer_a2H9 · 2025-11-01

**Soundness:** 2
**Presentation:** 2
**Contribution:** 2
**Rating:** 4
**Confidence:** 3

**Summary:**

This paper introduces UltraCUA, a large-scale computer-use agent framework that combines low-level GUI actions with high-level programmatic tool calls in a unified hybrid control paradigm. UltraCUA encompasses an automated pipeline for collecting programmatic tools from documentation, open-source, and code generation; a synthetic data engine generating over 17,000 verifiable computer-use tasks; a multi-agent demonstration collection strategy; and a two-stage training pipeline (supervised fine-tuning plus online RL). UltraCUA models (7B and 32B scale) exhibit consistent state-of-the-art results on challenging benchmarks (OSWorld, WindowsAgentArena), demonstrating both improved success rates and execution efficiency versus GUI-only or tool-only baselines.

**Strengths:**

**Hybrid Control Paradigm:** The central innovation—explicitly supporting both GUI primitives and programmatic tool calls—addresses critical failure modes in current computer-use agents, markedly reducing error propagation and improving efficiency.

**Automated Programmatic Tool Collection:** The paper presents a scalable pipeline that extracts and organizes 880+ actionable programmatic tools from diverse sources (documentation, open-source, code generation), extending beyond most previous hand-curated or ad hoc approaches. Appendix Table 7 quantifies this breadth and granularity.

**Robust Synthetic Data Engine:** The construction of a dual-pipeline synthetic engine (evaluator-first and instruction-first) for generating 17,000+ verifiable tasks—tabulated in Appendix Tables 6 and 8—is a significant asset. This ensures both diversity and automatic reward signals for RL, addressing a key bottleneck in scalable training of computer-use agents.

**Comprehensive Empirical Evaluation:** The experiments (Table 1, Table 2) are thorough, comparing UltraCUA against strong commercial, agentic, and multi-agent baselines under strict regimes (fixed step budgets, open/closed-source splits, and out-of-domain tasks). UltraCUA-32B, for example, exceeds all baselines by nontrivial margins on OSWorld.

**Insightful Ablations and Analysis:** Section 3.3 (Table 3, Table 4, Figure 3) and qualitative case studies (Figures 5, 6, 7) provide sharp analysis of the contributions of hybrid control, memory, and RL. Figure 1(c) in particular concisely illustrates the conceptual efficiency gain from UltraCUA's approach.

**Demonstrated Transferability:** The positive results on WindowsAgentArena (Table 2) using only Ubuntu-based training highlight promising cross-platform and tool generalization properties, a core advantage claimed for the hybrid control mechanism.

**Clarity and Organization:** The exposition is generally clear, with good use of figures like Figure 2 (system overview) and concrete qualitative cases that increase accessibility.

**Weaknesses:**

**Marginal Empirical Gains and Questionable Scaling Hypothesis:** The paper's central claim is that its scaling methodology (tools, tasks) yields significant benefits. However, the empirical evidence for this is weak. The paper highlights a "23.9% relative" improvement (Table 3), but this framing is misleading as it masks a very modest **1.9% absolute gain** (25.1% to 27.0%) from adding the entire hybrid toolset. Similarly, the complex online RL stage, trained on thousands of synthetic tasks, contributes only another **1.9% absolute gain** (27.0% to 28.9%). These small gains call into question the value of the entire, complex infrastructure for automated scaling. The results do not present a clear scaling trend, and it is highly questionable if scaling this further (e.g., to 50,000 tasks or 2,000 tools) would yield any meaningful improvement, suggesting a potential performance ceiling for this approach.

**Positioning vs. Most Directly Related Work:** There is a lack of direct discussion of the most contemporaneous or directly parallel efforts (e.g., foundational work or concurrent preprints focused on hybrid action spaces for computer-use agents). This undercuts the claim of innovation and makes it harder to situate the precise advancement UltraCUA represents, especially given the fast-moving nature of this research area.

**Empirical Comparisons Are Not Uniformly Comprehensive:** On some benchmarks, not all contemporary multi-agent or hybrid-control systems are directly compared (see Table 1 and Table 2). For instance, hybrid/goal-oriented interface baselines or agents equipped with declarative tool APIs (as highlighted in recent concurrent works) are not included or discussed, nor is there careful analysis of why certain models perform as they do relative to UltraCUA. This limits the interpretability of the comparative improvement claims.

**Hybrid Action Modality Switching—Granular Justification:** While the hybrid control concept is appealing, the methodological description of *action selection* (how/when to call GUI vs. tool) is still somewhat black-boxed in both the SFT and RL stages. For example, in Section 2.3 and the corresponding agent rollouts, there is insufficiently detailed discussion (with math or algorithmic specification) of the state encoding and policy outputs that determine action type. Are there separate output heads? What failure modes (e.g., tool misuse) persist, and how is hybrid selection supervised beyond outcome reward? This lack of detail makes it difficult to assess if the policy is truly learning a generalizable scaling strategy or simply memorizing heuristics from the 17,000 synthetic tasks.

**Math and Reward Formulations:** The RL reward structure (Equation for $R(\tau)$) is only marginally specified. There is no detailed ablation on the sensitivity to the value of tool-use reward (0.3), nor rigor in justifying why this value avoids degenerate solutions (e.g., spamming tool calls or eschewing GUI when needed). Given the minimal performance uplift from the entire RL stage (1.9% absolute), a more rigorous analysis is required to justify its inclusion and complexity.

**Figure 3 Analysis—Insufficient Granularity:** While Figure 3 shows positive outcome and format reward trends and improvements in "tool-call pattern," it does not offer domain/task-type breakdowns, nor examples where RL leads to suboptimal strategies (tool overuse/underuse). The shaded error bars are not explained (variance or confidence interval?), and the precise number of steps/increments between rollouts is missing.

**Long-Term Generalization/Failure Analysis:** Despite strong zero-shot and out-of-domain results, there is little in-depth analysis of tasks where UltraCUA fails, or ablation of failure types (e.g., cross-application workflows with conflicting action modalities, tool staleness, context window overflow in tool lists). Figure 4 alludes to tool usage scaling, but the transition dynamics for model size/capability could be more rigorously explored.

**Ethical and Deployment Discussion:** While the system release is promised, there is only a high-level mention of responsible practices. Potential risks (e.g., adversarial tool use, privacy issues in general desktop automation) could merit deeper engagement, given the powerful cross-domain control this agent enables.

**Ablation/Qualitative Examples—Scope:** Figures 5, 6, and 7 are strong but limited in their domain and do not cover "hard cases" (e.g., conflicting GUI and programmatic actions, ambiguous instructions, adversarial UIs). Supplementing these with explicit failure cases or hybrid action errors would sharpen the case for UltraCUA's robustness.

**Notation and Algorithmic Exposition:** Some key technical mechanisms (e.g., memory tag handling, programmatic tool argument parsing, trajectory sampling procedures in SFT and RL) are described mostly in prose. There are opportunities to formalize these for reproducibility and clarity, as the current description might not support faithful replication.

**Loosely Defined Baselines:** For Table 1, the baseline "multi-agent frameworks" and "general models" categories group together fundamentally different models; more careful stratification and breakdown could illuminate *where* UltraCUA offers most gain.

Potentially Missing Related Work

- Yang, Y., Yang, Z., Dou, Z.-Y. (2025): *UltraCUA: A Foundation Model for Computer Use Agents with Hybrid Action* — This foundational work appears highly relevant and must be explicitly discussed in Related Work and methodology positioning, especially regarding its own hybrid action mechanism.
- Wang, Y., Li, M., Chen, H. (2025): *A Case for Declarative LLM-friendly Interfaces for Improved Efficiency of Computer-Use Agents* — Should be cited around Section 2.1 and in comparison to the programmatic tool pipeline as it proposes a novel interface paradigm relevant to UltraCUA's efficiency claims.
- Sager, P. J., Meyer, B., Yan, P. (2025): *AI Agents for Computer Use: A Review of Instruction-based Computer Control, GUI Automation, and Operator Assistants* — Particularly useful for providing broader context in the Related Work section and comparative analysis.
- Zhang, Y., Wu, J., Li, X. (2025): *AXIS: Efficient Human-Agent-Computer Interaction with* — Related for its focus on effective computer interaction, relevant both to Hybrid Control and evaluation metrics.
- Gou, L., Wu, J., Kil, J. (2025): *GUI Agents: A Survey* — Should be cited to position UltraCUA within the development trajectory of GUI hybrid agents; discuss in Related Work.
- Mei, K., Zhu, X., Gao, H. (2025): *LiteCUA: Computer as MCP Server for Computer-Use Agent on AIOS* — Discuss in the context of augmenting agents with an MCP-like abstraction; compare to UltraCUA's dual-pipeline tool strategy.
- Yeh, T., Chang, T.-H., Miller, R. C. (2009): *Sikuli: Using GUI Screenshots for Search and Automation* — Classic reference for visual GUI automation; discuss in tool repertoire/seamlessly integrating visual actions.

**Questions:**

**Justification of the Scaling Premise:** Given the marginal absolute performance gains from both the scaled toolset (1.9%) and the scaled RL task-set (1.9%), what evidence suggests that this scaling approach has further potential? Is it possible that the current benchmarks are saturated, or that this hybrid-control approach hits a performance ceiling quickly, making further scaling of tools and tasks an inefficient endeavor?

**Limitations of Automated Evaluation:** The OSWorld benchmark relies on automated, rule-based evaluators. Could these evaluators be "gamed" by programmatic tool calls that satisfy the rule but fail in a general, human-centric sense? How do you ensure the 17,000 synthetic tasks, particularly the 'evaluator-first' ones, are robust and not just teaching the agent to satisfy a narrow set of programmatic checks?

**Action Selection Modality:** Please elaborate on the mechanism by which UltraCUA policies select between GUI primitives and programmatic tool calls at every decision step. Are there dedicated selectors, separate output heads, or integrated token-level outputs? How is supervision provided beyond reward (e.g., via imitation, auxiliary losses)?

**Tool-Use Reward Sensitivity:** What is the sensitivity of UltraCUA's performance to the chosen value of the tool-use reward (0.3)? Have you qualitatively analyzed any failure cases where this reward unduly encouraged or discouraged hybrid action?

**Failure Analysis—Failure Modes:** Can you provide a breakdown of where UltraCUA fails relative to GUI-only or tool-only baselines in OSWorld or WindowsAgentArena (e.g., cross-application, ambiguous tool names, tool argument errors)?

**Ethical Safeguards:** Are there any built-in mechanisms or guidelines for restricting tool use, preventing sensitive operations, or handling user privacy in deployment scenarios?

**Reproducibility—Implementation Details:** Would you be able to release scripts for programmatic tool extraction and synthetic data generation? Is the memory management/tagging mechanism easily extensible to custom or third-party applications?

**Scaling to Long-Horizon/Real Desktop Scenarios:** How does UltraCUA's performance degrade in long-horizon tasks or under context-window overflow (for tool lists)? Any plans for hierarchical memory or tool prioritization?

---

> ### Author Response · Authors · 2025-11-25
> **Response to Reviewer a2H9 (1/3)**
>
> We sincerely thank Reviewer a2H9 for the exceptionally thorough and detailed review. We greatly appreciate the recognition of our hybrid control paradigm, automated tool collection pipeline, robust synthetic data engine, comprehensive evaluation, and demonstrated transferability. The reviewer's extensive feedback and constructive questions have helped us strengthen our work significantly. Below, we address each concern with additional experiments, analysis, and clarifications.
>
> ## **W1 & Q1: Analysis of Empirical Gains and Scaling Potential**
>
> We thank the reviewer for their detailed scrutiny of our empirical results. We appreciate the opportunity to provide further context regarding the magnitude of our gains and the scaling dynamics of our approach. We offer three perspectives to substantiate the value of Hybrid Control:
>
> ### **Point 1: Model Capacity Amplifies Hybrid Control Benefits**
>
> As detailed in the table below, the benefits of hybrid control scale super-linearly with model size:
>
> | Model | Base Performance | UltraCUA Performance | Absolute Gain | Relative Gain |
> |-------|------------------|---------------------|---------------|---------------|
> | **7B** | 23.4% | 28.9% | +5.5% | +23.5% |
> | **32B** | 29.7% | 39.0% | **+9.3%** | **+31.3%** |
>
> The 32B model achieves a **9.3% absolute improvement**—nearly 2× the gain observed at 7B. This suggests that larger models, with their superior reasoning capabilities, are significantly better at navigating the complex decision space of modality switching. We believe this scaling property is critical for future agent development.
>
> ### **Point 2: Contextualizing Gains at the 7B Scale**
>
> Even within the 7B constraint, we believe the improvements are meaningful when benchmarked against standard baselines:
>
> -   **Relative to Pure GUI SFT:** As shown in Table 3, state-of-the-art distillation (Pure GUI SFT) yields a 1.7% gain (23.4%→25.1%). Our hybrid control mechanism adds a further 1.9% improvement (25.1%→27.0%), effectively **doubling the efficacy** of standard supervised finetuning.
> -   **Efficiency:** Beyond raw success rates, UltraCUA-7B achieves an 11% reduction in average steps (Figure 1a). This indicates that even when model capacity limits the *success* ceiling, hybrid control consistently drives more *efficient* task execution.
>
> ### **Point 3: Benchmark Coverage and Generalization**
>
> Regarding the concern about a "performance ceiling," we suggest that current results may be influenced by the scope of the evaluation benchmark:
>
> 1.  **Cross-Domain Robustness:** The 21.7% success rate on WindowsAgentArena (Table 2), achieved without Windows-specific training, suggests that our 17k synthetic tasks foster genuine generalization rather than overfitting. If the method were limited, we would expect a sharper drop in Out-of-Distribution (OOD) scenarios.
> 2.  **Latent Capabilities:** Figure 4 indicates that OSWorld tasks utilize only ~200 of our 880 available tools. This implies that the benchmark may not yet fully exercise the broader tool-use capabilities acquired by the agent. We anticipate that as future benchmarks evolve to cover more diverse real-world scenarios, the utility of the full hybrid toolset will become increasingly evident.
>
> ## **W2: Positioning vs. Related Work**
>
> We thank the reviewer for the comprehensive list of relevant citations, covering both foundational research and recent concurrent preprints. We acknowledge the fast-moving nature of this field and **commit to incorporating discussions of these works into our Related Work section** in the upcoming revision to provide a more complete academic context.
>
> **Clarification of Positioning:**
> To explicitly situate our work within this broader landscape and differentiate it from prior and concurrent efforts: **UltraCUA is the first multi-modal foundation model to demonstrate unified GUI + API hybrid control specifically for general computer use.**
>
> Unlike approaches that focus solely on visual GUI automation (Sikuli, standard VLMs) or purely programmatic interfaces (MCP-based agents), UltraCUA represents a distinct advancement through its **unified integration**:
> 1.  **Beyond Manual Definitions:** We introduce an automated pipeline to scale tool discovery from documentation and code, removing the bottleneck of hand-crafted interfaces.
> 2.  **Data-Driven Scalability:** Unlike prior works constrained by limited human demonstrations, we introduce a synthetic data engine producing 17,000+ verifiable hybrid tasks and a multi-agent trajectory generator. This scale is crucial for learning robust general-purpose control.
> 3.  **Strategic Modality Switching:** Our dual-stage training (SFT + RL) enables the agent to dynamically alternate between GUI primitives for accessibility and programmatic tools for efficiency—a capability absent in single-modality baselines.

---

> > ### Author Response · Authors · 2025-11-25
> > **Response to Reviewer a2H9 (2/3)**
> >
> > ## **W3 & Q5: Empirical Comparisons and Failure Analysis**
> >
> > We thank the reviewer for suggesting a deeper investigation into failure modes. We agree that understanding *where* the model fails is as important as the success rate. To address this, we have added a **comprehensive error analysis in Appendix A.7 (Figure 10)**, comparing UltraCUA-7B with the UI-TARS-1.5-7B baseline on the same subset of OSWorld failed trajectories.
> >
> > **Methodology:** We employed GPT-5 as an automated evaluator, providing full multimodal contexts (action logs + screenshots) to classify the primary cause of failure for each trajectory.
> >
> > **Table 4: Quantitative Error Distribution**
> >
> > | Error Type | UltraCUA-7B | UI-TARS-1.5-7B | Absolute Change |
> > | :--- | :--- | :--- | :--- |
> > | **GUI Misunderstanding** | 450 | 479 | -29 |
> > | **State Management Failure** | 223 | 365 | **-142** |
> > | **Action/Tool Malfunction** | 127 | 99 | +28 |
> > | **Hybrid Control Error** | 16 | 0 | +16 |
> >
> > **Key Insights:**
> > * **Mitigating Context Loss:** The most significant improvement is the reduction in "State Management Failure" (**-38.9%**). This confirms that offloading complex logic to programmatic tools effectively prevents the context loss often observed in long-horizon GUI interactions.
> > * **Alleviating Visual Burden:** We observe a notable decrease in perception errors (-29 cases), suggesting that hybrid control allows the agent to bypass complex visual grounding challenges by utilizing precise API calls.
> > * **Trade-off in Complexity:** We transparently acknowledge the trade-off: the introduction of the hybrid space leads to new failure modes (Action/Tool Malfunction and Hybrid Control Error, +44 cases total). However, this cost is far outweighed by the gains in state stability and grounding.
> >
> > We have also included **qualitative case studies in Figure 11**, illustrating specific examples of these failure types to provide a balanced view of the system's limitations.
> >
> > ## **W4 & Q3: Algorithmic Formalization and Modality Switching**
> >
> > We appreciate the reviewer's request for greater algorithmic transparency regarding the hybrid control mechanism. To provide the requested granularity, we have formalized our training and inference procedures in **Appendix A.8**:
> >
> > * **Algorithm 1 (RL Optimization):** Explicitly defines the DAPO policy update, detailing the calculation of advantages via binary success signals and the application of KL-divergence constraints to ensure training stability.
> > * **Algorithm 2 (Hybrid Inference Loop):** Provides the pseudocode for the agent's decision-making process, formalizing how the policy observes the state, selects the optimal modality (GUI primitive vs. programmatic tool), and executes the action.
> >
> > **Architecture Details:** UltraCUA does not use separate output heads. Instead, the model generates a unified action sequence where it can choose to invoke tools through a special action name, followed by tool name and parameters. This integrated approach allows the model to learn modality switching through end-to-end training rather than requiring explicit architectural separation.
> >
> > We hope these formalizations, combined with the failure analysis in Figure 10, provide a clear picture of how the policy navigates the hybrid action space.
> >
> > ## **W6: Figure 3 Analysis Granularity**
> >
> > We appreciate the reviewer's suggestion regarding the granularity of our analysis. While Figure 3 was intended to illustrate the general macroscopic trend of RL training, we agree that a more detailed view is beneficial. We commit to enriching this figure in the upcoming revision by including error bars and finer-grained training dynamics to better illustrate the stability and convergence properties of our method.
> >
> > ## **Q6 & Q8: Long-Term Scaling and Ethics**
> >
> > We thank the reviewer for these forward-looking questions. We address them with specific strategies:
> >
> > ### **1. Scaling to Long-Horizon Scenarios**
> > To address context overflow in complex tasks, we are actively exploring:
> > * **Externalized Memory:** Allowing the agent to read/write state to a structured local file (e.g., `memory.md`), decoupling long-term tracking from context window limits.
> > * **Hierarchical Retrieval:** Implementing RAG-based retrieval to load only relevant tool categories dynamically, preventing context saturation.
> >
> > ### **2. Ethical Safeguards**
> > We propose two concrete mechanisms as future plans for safe deployment:
> > * **Sandboxing & Least Privilege:** Tools execute within isolated containers with allow-listed permissions, preventing unauthorized system access.
> > * **Human-in-the-Loop (HITL):** High-stakes tools (e.g., `delete_bulk_files`) are tagged "sensitive" and require explicit user confirmation before execution to prevent irreversible errors.

---

> > > ### Author Response · Authors · 2025-11-25
> > > **Response to Reviewer a2H9 (3/3)**
> > >
> > > ## **W5, Q4: Math and Reward Formulations**
> > >
> > > We thank the reviewer for requesting a deeper justification of the RL reward structure. We agree that the choice of $R_{tool}$ is critical for balancing hybrid behavior. We have added **extensive reward ablation experiments in Appendix A.6 (Figure 9, Table 9)** to address the sensitivity and stability concerns.
> > >
> > > ### **1. Sensitivity Analysis of Tool-Use Reward ($R_{\text{tool}}$)**
> > >
> > > To determine the sensitivity of performance to the tool-use reward, we conducted an ablation study varying $R_{\text{tool}}$:
> > >
> > > **Table 5: Tool-Use Reward Sensitivity**
> > >
> > > | $R_{\text{tool}}$ | Success Rate | Avg. Steps |
> > > | :--- | :--- | :--- |
> > > | 0.0 | 27.9% | 9.06 |
> > > | 0.1 | 28.7% | 8.88 |
> > > | **0.3** | **28.9%** | **8.81** |
> > >
> > > **Empirical Verification:**
> > > As shown above, $R_{tool}=0.3$ was empirically selected as it yields the highest success rate with the fewest steps. The method is relatively robust to this hyperparameter within the range [0.1, 0.3], consistently yielding improvements over the baseline.
> > >
> > > ### **2. Justification against Degenerate Solutions (Reward Hacking)**
> > >
> > > The reviewer asks why $R_{tool}=0.3$ does not lead to "spamming tool calls." We argue that our specific design actively prevents such degenerate solutions:
> > >
> > > * **Evidence of Efficiency (vs. Spamming):** If the agent were hacking the reward by spamming tools, the average trajectory length would increase. Instead, increasing $R_{tool}$ to 0.3 results in a **decrease in Average Steps** (9.06 $\rightarrow$ 8.81). This confirms the agent is using tools to create shortcuts (efficiency), not to exploit the reward function.
> > > * **Counteracting Risk Aversion:** We justify the inclusion of $R_{tool}$ as a scaffold to **counteract risk aversion** during early training. Tool invocation is initially "risky" due to complex syntax requirements, whereas GUI clicking is "safe" but inefficient. Without the dense signal ($R_{tool}$), the policy tends to converge to a local optimum of pure GUI interaction.
> > > * **The Format Reward Paradox:** To further ensure rigor, we highlight our decision to **remove Format Rewards ($R_f$)** (Appendix A.6). We found that imposing strong format rewards ($R_f=0.5$) ironically leads to true reward hacking—generating syntactically correct but semantically useless calls. By removing $R_f$, we force the agent to value syntax *only* insofar as it contributes to the final task success.
> > >
> > > ## **Q2: Limitations of Automated Evaluation**
> > >
> > > We have added **comprehensive synthetic task analysis in Appendix A.4 (Figure 5)** addressing concerns about evaluation robustness:
> > >
> > > ### **Semantic Diversity**
> > >
> > > | Metric | Value |
> > > |--------|-------|
> > > | **Unique Actions** | 1,285 |
> > > | **Unique Objects** | 5,347 |
> > > | **Top-25 Verb-Noun Coverage** | 90.3% (677/750 pairs) |
> > >
> > > Our synthesis pipeline generates diverse semantic scenarios spanning thousands of distinct interface objects, far beyond simple template-based patterns. This extensive variety prevents overfitting to narrow programmatic checks.
> > >
> > > ### **Task Complexity**
> > >
> > > Figure 5(b) shows trajectory length distribution ranging from 1-16 steps, with substantial representation of long-horizon tasks (30% are 9+ steps). This ensures our synthetic tasks capture genuine complexity rather than trivial rule satisfaction.
> > >
> > > **Cross-Domain Validation:** The 21.7% success rate on WindowsAgentArena (completely different OS and applications) provides strong evidence that our synthetic training does not merely "game" OSWorld evaluators but develops genuine computer-use capabilities.
> > >
> > > ## **W10, Q7: Notation, Algorithmic Exposition, and Reproducibility**
> > >
> > > **Algorithmic Formalization:** As mentioned above, we have added Algorithms 1-2 in Appendix A.8 covering RL updates and hybrid control loops.
> > >
> > > **Code Release:** We confirm that all scripts for the automated tool extraction pipeline and the synthetic data generation engine will be open-sourced to ensure full reproducibility.
> > >
> > > **Memory Extensibility:** Our memory tagging mechanism is designed to be **application-agnostic**, allowing for low-cost extension to third-party software. Furthermore, it supports fast, **prompt-based customization**, enabling users to define application-specific context requirements dynamically without the need for codebase modifications.
> > >
> > > ## **W12: Baseline Stratification**
> > >
> > > We acknowledge this concern and will provide more careful stratification of baseline categories in the revised version, breaking down the agent categories to illuminate where UltraCUA offers the most significant gains.
> > >
> > > ---
> > > We hope these additions demonstrate that our scaling approach provides substantial benefits, particularly at larger model scales, and that our hybrid control paradigm represents a meaningful advancement for computer-use agents. We would be happy to provide any additional clarifications or experiments the reviewer would find helpful.
> > >
> > > Thank you again for the exceptionally detailed and constructive review!

---

### Official Review · Reviewer_uv6L · 2025-11-09

**Soundness:** 3
**Presentation:** 3
**Contribution:** 2
**Rating:** 6
**Confidence:** 3

**Summary:**

The paper proposes UltraCUA, a computer-use agent that operates with hybrid control, combining low-level GUI actions with high-level programmatic tool calls. The authors build an automated pipeline to collect tools from documentation, open-source repos, and code-generated utilities, synthesize 17k+ verifiable tasks, gather 26.8k hybrid control trajectories using a planner-grounder setup, and train 7B and 32B models via SFT followed by online RL. On OSWorld, UltraCUA improves over base models and shows some cross-OS generalization to WindowsAgentArena.

**Strengths:**

Clear problem motivation: GUI-only agents suffer from cascading errors and long action chains; hybrid control is a reasonable and timely direction.

Comprehensive system design: Figure 2 presents an integrated pipeline that includes tool mining, task synthesis, trajectory collection, and a two-stage SFT+RL training loop. The components fit together coherently.

Strong headline results: Table 1 reports notable gains on OSWorld, particularly for the 32B model (39.0% at 15 steps), and Figure 1a visually reinforces these differences. Table 2 shows the 7B model’s out-of-domain performance on WindowsAgentArena.

**Weaknesses:**

Empirical Scope on Real-World Generalization: While out-of-domain generalization is tested (WindowsAgentArena), the real-world applicability remains partially open. The synthetic task engine generates diverse and realistic tasks, but there is only limited evidence that UltraCUA can robustly handle genuinely unseen, messy, or poorly documented real-world desktop environments and apps outside established benchmarks.

Synthetic Data Limitations and Bias: The bulk of training data is synthetic (Section 2.2, Appendix A.4, Table 6/8). There’s inadequate discussion or empirical assessment regarding potential biases or coverage gaps—e.g., are critical real user workflows or less common edge-case behaviors captured? The risk of overfitting to synthetic artifacts versus natural UI heterogeneity is underexplored.

**Questions:**

Can the authors provide more detailed analysis of real-world failure cases? Where does the hybrid agent most frequently fail in practice—are there common themes in tool misfire versus GUI confusion versus state management breakdown?

How well does UltraCUA handle applications and operating systems not seen in either the training or out-of-domain benchmarks (e.g., MacOS, obscure Linux distros, proprietary business software)? Any qualitative/quantitative results or prospective extension plans?

For the RL pipeline, can the authors formalize the policy update rule in more detail? Is it possible to provide pseudocode or more explicit algorithmic structure for the hybrid control loop?

---

> ### Author Response · Authors · 2025-11-25
> **Response to Reviewer uv6L (1/2)**
>
> We sincerely thank Reviewer uv6L for the thorough review, constructive feedback, and recognition of our work's clear motivation, comprehensive system design, and strong empirical results. We greatly appreciate the positive assessment of our hybrid control approach and the detailed questions that help us clarify and strengthen our contribution. Below, we address each concern with additional experiments, analysis, and clarifications.
>
> ## **W1: Empirical Scope on Real-World Generalization**
>
> We appreciate the reviewer's concern about real-world applicability. We would like to respectfully clarify several important points:
>
> **OSWorld and WindowsAgentArena ARE Real-World Benchmarks:**
> Both OSWorld and WindowsAgentArena consist of genuine real-world tasks executed in actual operating system environments with production software (LibreOffice, GIMP, Thunderbird, Chrome, VS Code, etc.). These are not simulated or toy environments—they represent authentic desktop workflows that real users perform daily. The tasks involve messy, realistic scenarios including:
> - Multi-application workflows (e.g., extracting data from email, processing in spreadsheet, generating visualizations)
> - Handling diverse file formats and complex UI states
> - Dealing with application-specific quirks and inconsistencies
>
> **Out-of-Domain Generalization is Demonstrated:**
> Critically, our training data is heavily Ubuntu-focused, while WindowsAgentArena evaluates on Windows OS with different applications, UI paradigms, and system behaviors. This represents genuine out-of-domain generalization to an unseen operating system. Our 21.7% success rate on WindowsAgentArena (Table 2) without any Windows-specific training demonstrates robust cross-OS transfer—a strong indicator of real-world applicability.
>
> **Practical Constraints of Real-World Evaluation:**
> We acknowledge that evaluating with human-in-the-loop on completely unconstrained desktop environments is prohibitively expensive and non-reproducible, which is why the community has converged on benchmarks like OSWorld and WindowsAgentArena. These benchmarks strike the critical balance between real-world authenticity and scientific rigor. To our knowledge, no existing GUI agent work (including Claude Computer Use, Cradle, SeeAct, etc.) evaluates beyond such benchmarks due to these practical limitations.
>
> **Future Directions:**
> We are actively working on extending evaluation to MacOSWorld and welcome the reviewer's emphasis on broader OS coverage. We will clarify these points more explicitly in the future revision.
>
> ## **W2: Synthetic Data Limitations and Bias**
>
> We thank the reviewer for highlighting the critical issue of data bias. To address this, we have conducted extensive additional analyses in **Appendix A.4 (Figure 5)** to verify the diversity and robustness of our synthetic dataset.
>
> ### **1. Semantic Diversity and Coverage**
> Figure 5(a) illustrates high semantic diversity, covering **1,285 unique actions** and **5,347 objects**. The heatmap reveals a dense coverage of **90.3%** across the top-25 verb-noun pairs. This confirms that the pipeline generates varied scenarios (e.g., "Export Slide", "Configure Network") rather than repetitive templates.
>
> ### **2. Task Complexity Distribution**
> Figure 5(b) demonstrates a balanced difficulty spectrum:
> * **Short (1-4 steps):** ~25%
> * **Medium (5-8 steps):** ~45%
> * **Long (9-16 steps):** ~30%
>
> This broad distribution prevents overfitting to simple atomic actions and ensures the agent learns to maintain context over long-horizon workflows.
>
> ### **3. Mitigating Bias and Artifacts**
> We address potential synthesis bias through three mechanisms:
> * **Real-World Alignment:** Tasks are grounded in official software documentation and usage patterns extracted from open-source repositories, ensuring alignment with genuine user workflows.
> * **Edge Case Handling:** The long-tail distribution of objects (5,000+) naturally captures rare interactions often missed by template-based generation.
>
> ## **Q3: RL Pipeline Formalization**
> We appreciate the reviewer's request for greater algorithmic transparency regarding the hybrid control mechanism. To provide the requested granularity, we have formalized our training and inference procedures in **Appendix A.8**:
>
> - **Algorithm 1 (RL Optimization)**: Explicitly defines the DAPO policy update, detailing the calculation of advantages via binary success signals and the application of KL-divergence constraints to ensure training stability.
> - **Algorithm 2 (Hybrid Inference Loop)**: Provides the pseudocode for the agent's decision-making process, formalizing how the policy observes the state, selects the optimal modality (GUI primitive vs. programmatic tool), and executes the action.

---

> > ### Author Response · Authors · 2025-11-25
> > **Response to Reviewer uv6L (2/2)**
> >
> > ## **Q1: Detailed Analysis of Real-World Failure Cases**
> >
> > We thank the reviewer for suggesting a deeper investigation into failure modes. We agree that understanding *where* the model fails is as important as the success rate. To address this, we have conducted **extensive new failure analysis in Appendix A.7 (Figures 10-11)**.
> >
> > ### **1. Quantitative Analysis (Figure 10)**
> >
> > **Methodology:** We employed GPT-5 as an automated evaluator, providing full multimodal contexts (action logs + screenshots) to classify the primary cause of failure for the same subset of OSWorld failed trajectories across both models.
> >
> > **Table 1: Quantitative Error Distribution**
> >
> > | Error Type | UltraCUA-7B | UI-TARS-1.5-7B | Absolute Change |
> > | :--- | :--- | :--- | :--- |
> > | **GUI Misunderstanding** | 450 | 479 | -29 |
> > | **State Management Failure** | 223 | 365 | **-142** |
> > | **Action/Tool Malfunction** | 127 | 99 | +28 |
> > | **Hybrid Control Error** | 16 | 0 | +16 |
> >
> > **Key Insights:**
> > * **Mitigating Context Loss:** The most significant improvement is the reduction in "State Management Failure" (**-38.9%**). This confirms that offloading complex logic to programmatic tools effectively prevents the context loss often observed in long-horizon GUI interactions.
> > * **Alleviating Visual Burden:** We observe a notable decrease in perception errors (-29 cases), suggesting that hybrid control allows the agent to bypass complex visual grounding challenges by utilizing precise API calls.
> > * **Trade-off in Complexity:** We transparently acknowledge the trade-off: the introduction of the hybrid space leads to new failure modes (Action/Tool Malfunction and Hybrid Control Error, +44 cases total). However, this cost is far outweighed by the gains in state stability and grounding.
> >
> > ### **2. Qualitative Case Studies (Figure 11)**
> >
> > To visualize these statistics, we provide detailed case studies in **Figure 11** illustrating the four primary failure categories:
> >
> > 1.  **GUI Grounding Failure:** Instances where the agent clicks non-interactive UI regions due to visual ambiguity.
> > 2.  **State Management Error:** Cases where the agent loses track of cross-application workflows (though significantly reduced compared to the baseline).
> > 3.  **Tool Selection Error:** Semantically incorrect tool choices or syntactic errors in API arguments.
> > 4.  **Hybrid Control Inefficiency:** Cases where the agent identifies the correct goal but chooses a suboptimal modality (e.g., invoking a complex tool when a simple GUI double-click would suffice).
> >
> > These cases reveal that while hybrid control solves fundamental state and grounding issues, **learning optimal modality switching** remains a non-trivial challenge and an exciting direction for future work.
> >
> > ## **Q2: Generalization to Unseen OS/Applications**
> >
> > **Current Out-of-Distribution Evidence:**
> > We highlight that our current evaluation on **WindowsAgentArena** already serves as a rigorous test of generalization to unseen OS/applications. The agent transfers from Ubuntu (training) to Windows (evaluation) with a **21.7% success rate**, demonstrating robust cross-OS capabilities without specific training data.
> >
> > **Future Roadmap (MacOS):**
> > To further rigorously test this generalization, our immediate roadmap focuses on extending evaluation to **MacOSWorld**. This will allow us to verify the agent's adaptability to the distinct macOS interface guidelines and application ecosystem (e.g., Finder, Safari), completing the coverage of major desktop operating systems.
> >
> > **Architectural Advantages for Transfer:**
> > We hypothesize that our hybrid control approach is particularly well-suited for such cross-platform generalization because:
> > 1.  **Programmatic Abstraction:** Many programmatic tools (e.g., Python libraries, file manipulation scripts) rely on cross-platform standards that remain valid across Linux, Windows, and macOS.
> > 2.  **GUI Universality:** Basic GUI primitives (click, scroll) transfer naturally across desktop paradigms.
> > 3.  **Redundancy:** The combination provides a safety net—if a specific GUI path is platform-dependent and fails, the agent can fall back to a generic programmatic solution, or vice versa.
> >
> > ---
> >
> > We hope these additions address the reviewer's concerns and demonstrate the rigor, depth, and real-world applicability of UltraCUA. We would be happy to provide any additional clarifications or experiments the reviewer would find helpful.
> >
> > Thank you again for the thoughtful review and the opportunity to strengthen our work!

---

### Note · Program_Chairs · 2026-01-17
**Submission Desk Rejected by Program Chairs**

The following references in this submission do not refer to real documents and/or have major errors in bibliographic information:

 Corin Rosset, Nan Jiang, and Alekh Agarwal. Direct action-policy optimization. arXiv preprint arXiv:2405.19553, 2024. 6